# Synthesis of Temporin-SHa Retro Analogs with Lysine Addition/Substitution and Antibiotic Conjugation to Enhance Antibacterial, Antifungal, and Anticancer Activities

**DOI:** 10.3390/antibiotics13121213

**Published:** 2024-12-13

**Authors:** Shahzad Nazir, Arif Iftikhar Khan, Rukesh Maharjan, Sadiq Noor Khan, Muhammad Adnan Akram, Marc Maresca, Farooq-Ahmad Khan, Farzana Shaheen

**Affiliations:** 1Third World Center for Science and Technology, International Center for Chemical and Biological Sciences, University of Karachi, Karachi 75270, Pakistan; shahzad.nazir@iccs.edu (S.N.); arififtikhar@iccs.edu (A.I.K.); rukesh.maharjan@iccs.edu (R.M.); sadiqnoorkhan172@gmail.com (S.N.K.); adnan.akram@iccs.edu (M.A.A.); farooq.khan@iccs.edu (F.-A.K.); 2Aix Marseille Univ, CNRS, Centrale Med, ISM2, 13013 Marseille, France

**Keywords:** temporin-SHa, levofloxacin, antimicrobial peptides, AMPs, antibacterial, antifungal, anticancer

## Abstract

In the face of rising the threat of resistant pathogens, antimicrobial peptides (AMPs) offer a viable alternative to the current challenge due to their broad-spectrum activity. This study focuses on enhancing the efficacy of temporin-SHa derived NST-2 peptide (**1**), which is known for its antimicrobial and anticancer activities. We synthesized new analogs of **1** using three strategies, i.e., retro analog preparation, lysine addition/substitution, and levofloxacin conjugation. Analogs were tested in terms of their antibacterial, antifungal, and anticancer activities. Analog **2,** corresponding to retro analog of NST-2, was found to be more active but also more hemolytic, reducing its selectivity index and therapeutic potential. The addition of lysine (in analog **3**) and lysine substitution (in analog **7**) reduced the hemolytic effect resulting in safer peptides. Conjugation with levofloxacin on the lysine side chain (in analogs **4** and **5**) decreased the hemolytic effect but unfortunately also the antimicrobial and anticancer activities of the analogs. Oppositely, conjugation with levofloxacin at the N-terminus of the peptide via the β-alanine linker (in analogs **6** and **8**) increased their antimicrobial and anticancer activity but also their hemolytic effect, resulting in less safe/selective analogs. In conclusion, lysine addition/substitution and levofloxacin conjugation, at least at the N-terminal position through the β-alanine linker, were found to enhance the therapeutic potential of retro analogs of NST-2 whereas other modifications decreased the activity or increased the toxicity of the peptides.

## 1. Introduction

The rise in resistant pathogens has drawn attention to antimicrobial peptides (AMPs), which possess broad-spectrum efficacy. AMPs operate through a unique mechanism of action that makes it harder for pathogens to develop resistance [1]. Despite this potential, many challenges like instability, protease susceptibility, hemolysis, and toxicity have limited their clinical use. Researchers have explored several ingenious strategies to address these issues. For example, one early method involved swapping the natural L-amino acids with unnatural D-amino acids to makes these peptides invisible to proteases, which typically recognize and cleave natural amino acid chains. This seemingly minor tweak enhances the stability of AMPs in the bloodstream [2]. Another effective approach that has shown promise is peptide cyclization, which not only lowers the toxicity but also improves antimicrobial activity [3]. Researchers have also tried attaching polyethylene glycol (PEG) to peptides (PEGylation), wherein PEG acts as a hydrophilic shield to improve peptide solubility in aqueous environments. PEGylation was found to enhance the renal clearance of peptides, thereby preventing unwanted build-up to lower the potential side effects of AMPs [4,5]. Other modifications like glycosylation (attaching sugar molecules) or lipidation (conjugating fatty acids) have extended the functional activities of AMPs [6,7]. Additionally, flipping the order of amino acids to make retro analogs, and reversing the handedness of amino acids to create inverso-analogs, has been shown to enhance the potency of AMPs [8]. Taking these two concepts a step further, retro-inverso peptides were also developed, which are basically mirror images of the original molecule with a reversed amino acid sequence. This double twist alters the stereochemical and conformational properties of the peptides, potentially leading to new AMPs with beneficial properties [9,10]. The incorporation of non-canonical amino acids can significantly enhance the stability of the peptides and improve their biological functions [11,12]. These amino acids have also been utilized as secondary metabolites, thus performing physiological processes. Although their role is not fully understood yet, they hold great potential for optimizing the overall activity of peptides and molecules. Notable examples include seralasin for hypertension, icatibant for enhance stability, and carbetocin for postpartum hemorrhage [13,14]. Peptide analogs, known as peptidomimetics, are designed to mimic the structural elements of natural peptides to enhance biological functions and mitigate the drawbacks of natural peptides [15,16,17]. Notable examples include triazolyl peptidomimetics, which have shown promise as enzyme inhibitors as well as anticancer, antibacterial, and antifungal agents [18,19]. All these innovative strategies by researchers offer a glimpse of hope in improving the activities of AMPs, which can pave the way to developing effective antimicrobial therapies [20,21].

Conjugating heterocycles with small peptides is also known to improve their biological properties. Since the year 2000, numerous heterocyclic-conjugated peptides with potential therapeutic benefits have been reported [22,23,24,25,26,27,28,29,30]. Likewise, arene peptide conjugates were also developed with promising medicinal properties [31,32,33,34]. Antibiotic drugs like levofloxacin gained our attention due to the broad-spectrum antimicrobial effect. Recently, its conjugation with an antimicrobial peptide named indolicidin was reported by Ghaffar et al. [35]. Levofloxacin M33 peptide conjugate did not show any major change in its ability to fight Gram-negative bacteria [36]. Interestingly, when a linear peptide (R4W4) was conjugated with levofloxacin, an improvement of antibacterial activity with low hemolytic effect was observed [37,38]. Early research looked into retro analogs of P53 peptides using molecular dynamics simulations and circular dichroism spectroscopy [39,40]. Temporins are a diverse group of short AMPs, which were originally isolated from the skin of European frogs [41,42,43]. The founding members of this group of AMPs include temporin A, temporin B, and temporin L [44,45]. Temporins exhibit selective antibacterial activity against Gram-positive bacteria, with significantly lower efficacy against Gram-negative bacteria. Their reduced activity against Gram-negative strains is attributed to lipopolysaccharide (LPS)-induced aggregation in the outer membrane, which limits their antibacterial effectiveness [46,47,48,49,50,51]. Temporin-SHa has been well-documented for its strong antibacterial activity against Gram-positive bacteria, including *Staphylococcus aureus*, *Bacillus subtilis*, and *Enterococcus faecalis*, while exhibiting comparatively lower activity against Gram-negative bacteria, such as *Escherichia coli*, *Pseudomonas aeruginosa*, *Acinetobacter baumannii*, and *Klebsiella pneumoniae*. Similarly, D-analog substitutions of temporin-SHa have also demonstrated potent activity against Gram-positive bacteria but reduced efficacy against Gram-negative strains. This lower activity against Gram-negative bacteria is likely due to differences in membrane composition and permeability barriers, which are characteristic of Gram-negative pathogens. Consequently, the analogs presented in this manuscript also exhibit high MIC values against Gram-negative bacteria compared to Gram-positive counterparts [52,53,54]. Recently, we used the Fmoc peptide synthesis strategy to prepare temporin-SHa analogs, wherein natural glycine-10 residue of the parent peptide was replaced with atypical (2-naphthyl)-D-alanine, D-tyrosine and D-phenylalanine to study the effects of hydrophobic residues on antimicrobial efficacy [52]. Previously, we also synthesized [G4a]-SHa, [G7a]-SHa, and [G10a]-SHa analogs by substituting glycine residue with D-alanine at position 4, 7, and 10, respectively [55]. Among these analogs, NST-2, corresponding to the [G4a]-SHa peptide, demonstrated promising activity against methicillin-resistant *Staphylococcus aureus* (NCTC 13277) with an MIC of 14.3 μM. It also showed low hemolysis compared to the parent temporin-SHa peptide [56]. These results make this analog a suitable choice for further investigation [53].

The peptide analog NST-2 is a D-alanine modified variant of temporin-SHa [55]. In this work, we decided to synthesize NST-2 in a retro manner potentially increasing its activity [57]. Then, we further modified the retro analog of NST-2 by either adding lysine residue to its C terminus or by substitution of glycine at the 4th position of this retro analog with lysine. Finally, to potentially further enhance the therapeutic potential of the peptide, we conjugated the antibiotic levofloxacin to either the side chain of lysine-3 or lysine-14 of the retro analog or to the N-terminal phenylalanine-1 via a *β*-alanine linker. A summary of these modifications in the retro analog of NST-2 is provided in Figure 1.

## 2. Results

Lysine-enriched peptides are known to possess strong antimicrobial activities with less toxicity toward eukaryotic cells [58,59,60]. Hence, we decided to investigate the role of lysine using two approaches—addition and substitution—in the retro analog of NST-2 peptide. Briefly, L-lysine was added to retro analog **2** at position-14, which afforded RSP-1 peptide (**3**). In addition, glycine at position-4 in the retro analog was substituted with lysine to give RSP-4 peptide (**7**). In the next phase, antibacterial levofloxacin was selected for direct conjugation with the side chain of lysine-14 and lysine-3 of analog **3** to afford RLFP-1 (**4**) and RLPF-2 (**5**), respectively. Similarly, levofloxacin was also conjugated to analog **3** and analog **7** at phenylalanine-1 via β-alanine linker to obtain RLFP-3 (**6**) and RLFP-4 (**8**), respectively. Amino acid sequences of the peptides (**1**–**8**) are shown in Table 1. All the peptides were purified with RP-HPLC (PuriFlash^®^, Interchim, Montluçon, France) using the PFB15C18XS-250/212 column, and eluted at a flow rate of 3 mL/min by 0.1% TFA in H_2_O/ACN (40:60). Their purity was established by UPLC (Agilent 1260 Infinity Diode Array, C-4 reversed-phase analytical column 5 μm, 150 × 4.6 mm, Santa Clara, CA, USA). Furthermore, 1D/2D NMR, UV–Vis., FT-IR spectroscopy, polarimetry, and HR-MS-ESI mass spectrometry were employed to characterize the peptides. Physiochemical parameters of the synthesized peptides showing the molecular weights, optical rotation and retention time are shown in Table 2.

### 2.1. Circular Dichroism (CD) and Secondary Structure Analysis

Sodium dodecyl sulfate (SDS) is commonly used to denature proteins as it disrupts non-covalent interactions. Peptides can form β-sheets depending on their hydrophobic residues, resistance to SDS-induced denaturation, helical content, and interactions with SDS micelles [61,62]. Antiparallel refers to one of the orientations of β-sheets, wherein the strands align in opposite directions. This arrangement, along with parallel and twisted configurations, adds to the variety of structural patterns that can be observed in spectral analyses. For precise and detailed structural information derived from CD spectra, the Beta Structure Selection (BeStSel) method has become a widely utilized tool nowadays [63,64]. The circular dichroism (CD) spectra of temporin NST-2 (**1**) and its analogs in 20 mM SDS are displayed in Figure 1. Spectra show that in lipid-like environments provided by SDS, all analogs adopt an alpha-helical conformation due to their amphipathic properties, which is crucial for their antimicrobial function by facilitating their insertion into bacterial membranes. The hydrophobic segment of the peptide interacts with the lipid’s hydrophobic portion, while the hydrophilic segment remains accessible to the aqueous environment. The percentage of alpha-helix and other secondary structures were quantified from CD data using Bestsel software (https://bestsel.elte.hu/index.php, accessed on 22 October 2024) and are shown in Table 3. NST-2 (**1**) exhibited an 81.5% alpha-helical content, while its retro analog RNST (**2**) displayed a 62% alpha-helical structure, showing that the inversing amino acid sequence had a major effect on the secondary structure of NST-2. The addition of Lys at the C-terminus in analog **3** restored the percentage of the alpha-helix to 81.8% whereas Lys substitution at position 4 in analog **7** decreased further helix content to 56.8%. Conjugation with levofloxacin on Lys^3^ (analog **4**) or Lys^14^ (analog **5**) also restored the helix content to 80.1 and 80.4%, respectively. Conjugation with levofloxacin on Phe^1^ (analogs **6** and **8**) also increased the alpha-helical content compared to analog **2** with 82.2 and 75.0%, respectively.

**Table 1 antibiotics-13-01213-t001:** Amino acid sequence of peptides (**1**–**8**) with changes shown in red color (R = reverse sequence, LF = levofloxacin, β-Ala = beta alanine).

Peptide Name	Systematic Name	Sequence
NST-2 (**1**)	[G4a]-SHa	H-Phe^1^-Leu^2^-Ser^3^-D-Ala^4^-Ile^5^-Val^6^-Gly^7^-Met^8^-Leu^9^-Gly^10^-Lys^11^-Leu^12^-Phe^13^-NH_2_
RNST-2 (**2**)	R[G4a]-SHa	H-Phe^1^-Leu^2^-Lys^3^-Gly^4^-Leu^5^-Met^6^-Gly^7^-Val^8^-Ile^9^-D-Ala^10^-Ser^11^-Leu^12^-Phe^13^-NH_2_
RSP-1 (**3**)	RNST-2-14K	H-Phe^1^-Leu^2^-Lys^3^-Gly^4^-Leu^5^-Met^6^-Gly^7^-Val^8^-Ile^9^-D-Ala^10^-Ser^11^-Leu^12^-Phe^13^-Lys^14^-NH_2_
RLFP-1 (**4**)	RRNST-2-14K-14LF	H-Phe^1^-Leu^2^-Lys^3^-Gly^4^-Leu^5^-Met^6^-Gly^7^-Val^8^-Ile^9^-D-Ala^10^-Ser^11^-Leu^12^-Phe^13^-Lys^14^(LF)-NH_2_
RLFP-2 (**5**)	RNST-2-14K-3LF	H-Phe^1^-Leu^2^-Lys^3^(LF)-Gly^4^-Leu^5^-Met^6^-Gly^7^-Val^8^-Ile^9^-D-Ala^10^-Ser^11^-Leu^12^-Phe^13^-Lys^14^-NH_2_
RLFP-3 (**6**)	RNST-2-14K-1LF	LF-β-Ala-Phe^1^-Leu^2^-Lys^3^-Gly^4^-Leu^5^-Met^6^-Gly^7^-Val^8^-Ile^9^-D-Ala^10^-Ser^11^-Leu^12^-Phe^13^-Lys^14^-NH_2_
RSP-4 (**7**)	RNST-2-G4K	H-Phe^1^-Leu^2^-Lys^3^-Lys^4^-Leu^5^-Met^6^-Gly^7^-Val^8^-Ile^9^-D-Ala^10^-Ser^11^-Leu^12^-Phe^13^-NH_2_
RLFP-4 (**8**)	RNST-2-G4K-1LF	LF-β-Ala-Phe^1^-Leu^2^-Lys^3^-Lys^4^-Leu^5^-Met^6^-Gly^7^-Val^8^-Ile^9^-D-Ala^10^-Ser^11^-Leu^12^-Phe^13^-NH_2_

**Table 2 antibiotics-13-01213-t002:** Physiochemical parameters of the synthesized peptide analogs showing molecular weights, optical rotation, and retention time of each compound obtained from UPLC.

Peptide Name	Chemical Formula	Exact Mass	Observed Mass *	Time_(R)_ **	${\left[\alpha \right]}_{D}^{25}$ ^‡^	Yield ^§^
RNST-2 (**2**)	C_68_H_111_N_15_O_14_S	1393.8	1395.8 [M + H]^+^	3.1	−202	25
RSP-1 (**3**)	C_74_H_123_N_17_O_15_S	1521.9	1523.9 [M + H]^+^	2.9	−15	8
RLFP-1 (**4**)	C_92_H_141_FN_20_O_18_S	1865.0	1863.5 [M + H]^+^	3.3	+120	22
RLFP-2 (**5**)	C_92_H_141_FN_20_O_18_S	1865.0	1912.3 [M + 2Na]^2+^	3.7	+5	17
RLFP-3 (**6**)	C_95_H_146_FN_21_O_19_S	1936.0	1936.9 [M + H]^+^	3.6	−45	26
RSP-4 (**7**)	C_72_H_120_N_16_O_14_S	1464.9	1467.0 [M + H]^+^	3.1	+88	28
RLFP-4 (**8**)	C_93_H_143_FN_20_O_18_S	1879.1	941.7 [M + 2H]^2+^	3.0	+131	31

* Via ESI-MS; ** retention time in minutes; ^‡^ recorded in MeOH; ^§^ overall % yield.

**Table 3 antibiotics-13-01213-t003:** Percentage of the different types of secondary structures determined from CD data using Bestsel (https://bestsel.elte.hu/index.php, accessed on 22 October 2024).

Peptide	Helix (%)	Antiparallel (%)	Parallel (%)	Turn (%)	Others (%)
NST-2 (**1**)	81.5	18.5	0.00	0.00	0.00
RNST (**2**)	62.0	17.9	20.1	0.00	0.00
RSP-1 (**3**)	81.8	18.2	0.00	0.00	0.00
RLFP-1 (**4**)	80.1	19.9	0.00	0.00	0.00
RLFP-2 (**5**)	80.4	19.6	0.00	0.00	0.00
RLFP-3 (**6**)	82.2	17.8	0.00	0.00	0.00
RSP-4 (**7**)	56.8	22.1	0.00	7.6	13.6
RLFP-4 (**8**)	75.0	0.00	14.1	0.00	11.0

### 2.2. Antimicrobial Assay

NST-2 (**1**) has been already reported to possess strong antibacterial activity against *Helicobacter pylori* (ATCC 43504) and *Staphylococcus aureus* (NCTC 13277) with low hemolytic effect [55,56]. The retro analog of **1**, RNST-2 (**2**), was remarkably active against *S. aureus* (NCTC 13277), *B. subtilis* (ATCC 23857), *S. typhi* (ATCC 14028), *E. coli* (ATCC 25922), and *P. aeruginosa* (ATCC 10145) as well as against *Candida albicans* (ATCC 36082) (Table 4) with MIC values lower than the ones of **1** in all cases. Similarly, RSP-1 peptide (**3**) obtained by addition of Lys at the C-terminal end of analog **2** also showed higher antimicrobial activities compared to **1**. Further modifications were explored by synthesizing RSP-4 (**7**), an analog of **2,** wherein Gly^4^ was substituted with Lys. Although supposed to increase it, this substitution did not improve the antibacterial activity of the analog and rather inhibited it as evident from higher MIC values compared to **2**. The antibacterial molecule levofloxacin was then conjugated to some analogs to try to improve their activity. As expected, the addition of levofloxacin at position 1 in analog **7** via the β-alanine linker generated RLFP-4 (**8**) with an improved activity against both Gram-positive and -negative bacteria, but not against C. albicans. Similarly, RLFP-3 (**6**), wherein levofloxacin was again conjugated to Phe^1^ of analog **3,** also via the β-alanine linker, demonstrated potent activities against both Gram-positive and -negative bacteria. Surprisingly, and not expected due to the selective antibacteria activity of levofloxacin, it was also the most active analog against *C. albicans* with an MIC of 3.12 µM. In contrast to analogs **6** and **8**, RLFP-1 (**4**) and RLFP-2 (**5**) obtained by conjugation with levofloxacin to the side chain of Lys^14^ or Lys^3^ of analog **3** displayed a reduced antimicrobial effect compared to parent analog **3**.

To further explore the mechanism of action of the analogs, three bacterial strains (*S. typhi*, *E. coli*, and *P. aeruginosa*) were treated with the most potent peptide analogs and analyzed through the AFM technique in tapping mode. Figure 2A–D, Figure 3A–D, and Figure 4A–D represent the *S. typhi*, *E. coli*, and *P. aeruginosa* bacteria, respectively. Untreated *S. typhi* (Figure 2A) appeared as short rods ranging its length between 0.8 and 1.5 µm, and a slightly irregular texture with no signs of damage or disruption. *S. typhi* treated with RLFP-4 (**8**), RLFP-3 (**6**), and RLFP-1 (**4**) at 2XMIC showed a loss of morphological rod shapes. The loss of structural integrity could be seen due to leakage of cytoplasmic contents around the damaged cells demonstrating that the analogs caused membrane damage. Similarly, treatment of *E. coli* (Figure 3) or *P. aeruginosa* (Figure 4) with RLFP-3 (**6**), RSP-1 (**3**), and RLFP-4 (**8**) at 2XMIC values caused significant damages to the bacteria as their rod-shaped morphologies were lost, cells were disintegrated, and their cytoplasmic content were found around some rod-shaped cells confirming again the membrane damages caused by the analogs.

### 2.3. Anticancer Activity and Hemolytic Effect of the Analogs

NST-2 (**1**) was previously reported with anti-breast cancer activity (IC_50_: 17.5 µM) in MCF-7 cells, while it was inactive against cervical cancer HeLa cells [65] with a hemolysis HC_50_ value of 90.0 µM [56]. The antiproliferative activity of the retro analog of NST-2 against cancer cells, i.e., MCF-7 and HeLa cells, as well as their hemolytic properties are presented in Figure 5. IC_50_ and HC_50_ values were determined from Figure 5 and are reported in Table 5. Compared to parent peptide NST-2 (**1**), retro analog **2** was found less active on MCF-7 (Figure 5A) (IC_50_ of 17.9 versus 53.0 µM) but more active on HeLa (Figure 5B) (IC_50_ of >100 versus 60.0 µM). Compared to parent analog **2**, the analogs **3**, **6**, **7**, and **8** in which Lys was added were found to be more active than **2** on MCF-7 cells with an efficiency order of **6** > **8** > **7** > **3** > **2**. On HeLa cells, analogs **3**, **6**, **7**, and **8** were found to be more active than analog **2** with an efficiency order of **6** > **8** > **3** > **7** > **2**. Analogs **4** and **5,** in which levofloxacin was added to the side chain of Lys^14^ or Lys^3^, were found inactive with IC_50_ > 100 µM. Interestingly, analog **6** was the only analog with an IC_50_ on MCF-7 cells lower than the one of parent peptide NST-2 (**1**) (i.e., 13.3 versus 17.9 µM, respectively) and was also the more active on HeLa cells (i.e., 12.1 µM).

In terms of hemolysis (Figure 5C and Table 5), all analogs were found to be more hemolytic than analog **1** (with HC_50_ values ranging from 4.5 to 51.0 µM compared to 90.0 µM) except analog **4** with a similar HC_50_ (i.e., 98.9 µM) and analog **7** that was found to not be hemolytic at the tested concentrations (HC_50_ > 100 µM). When compared with their parent molecule (i.e., the analog **2**), analogs **3**, **4**, **5**, and **7** were found to be less hemolytic than analog **2** (HC_50_ values of 51.0, 98.9, 25.0, and >100 µM versus 13.8 µM) confirming that the addition of Lys to AMP sequences reduces their hemolytic effect. Oppositely, analogs **6** and **8,** in which levofloxacin was added at position 1, were found to be more hemolytic than **2** (HC_50_ of 6.6 and 4.5 µM versus 13.8 µM, respectively).

### 2.4. Selectivity Indexes of the Analogs

The selectivity of action of the analogs against micro-organisms or cancer cells was further evaluated through calculation of their selectivity indexes (SIs). SIs were determined using either the MIC on micro-organisms or IC_50_ on cancer cells compared to HC_50_ values, and they are reported in Table 6.

Regarding the SIs on the micro-organisms, it appears that the retro analog **2** had a lower SI compared to analog **1** (i.e., 8.8 versus 14.4). When compared to parent analog **2**, all the other analogs derived from **2** were also found less selective (due to higher MIC values and/or lower HC_50_), except analogs **3** and **7**, for which the SIs on the micro-organisms were equal (for **7**) or superior (3.7-fold increase for **3**) to parent analog **2**. This confirmed that addition of Lys into AMPs improves their selectivity through increased antimicrobial activity and/or decreased toxicity toward human cells, this being the case for analog **3** with a decreased toxicity (HC_50_ of 13.8 versus 51.0 µM for analogs **2** and **3**) and a conserved antimicrobial activity (lowest MIC of 1.56 µM for analogs **2** and **3**). Regarding levofloxacin conjugate analogs, their selectivity was reduced either due to a decrease in antimicrobial effect or to increase in their toxicity. Analog **4** corresponding to the levofloxacin conjugate on Lys^14^ displayed a reduced toxicity (HC_50_ of 98.6 µM versus 13.8 µM for analog **2**) but also, unfortunately reduced antimicrobial activity (lowest MIC of 25 µM for **4** versus 1.56 µM for **2**), resulting in a SI of 3.9 versus 8.8. Analogs **6** and **8** corresponding to the levofloxacin conjugates at position 1 were as active as analog **2** (lowest MIC of 1.56 µM) but their higher toxicity (HC_50_ of 6.6 and 4.5 µM versus 13.8 µM) reduced their selectivity from 8.8 to 4.2 and 2.8, respectively. Regarding the SIs on cancer cells, all analogs were less selective than analog **1**, but when compared to analog **2** from which they originated, analogs **3**, **6**, and **7** were found to be more selective, suggesting again that the presence of additional Lys in the sequence of retro NST-2 also improves its selectivity against cancer cells.

## 3. Discussion

In the present study, retro analogs of NST-2, a D-alanine variant of temporin-Sha, were synthetized and tested in terms of their antimicrobial and anticancer activities. This strategy was based on the fact that retro analogs of peptides have been described in the literature as more active than their parent peptides [8,9,10,20]. Retro analogs were further modified by the addition of lysine residue, a strategy also known to improve AMPs activity and to decrease their hemolytic effect [58,59,60]. Incorporation of cationic lysine residue reduces the aggregation process in an aqueous environment, thereby enhancing the oligomerization as it binds to lipopolysaccharides (LPSs). This interaction results in the disruption of LPS aggregates, and helps to neutralize the LPS-induced inflammation in animal models, including mice and rats [66]. Furthermore, the conjugation of an LPS-binding motif to an antimicrobial peptide (AMP) can effectively neutralize endotoxins and disrupt the bacterial outer cell wall. This motif, referred to as the “boomerang motif”, represents a promising strategy for designing cell wall permeabilizing peptides [67]. Finally, retro analogs were conjugated to levofloxacin (either on the side chain of Lys or at the N-terminus of the peptide); the literature describes that AMPs conjugated to antibiotics displaying, or not, increased efficiency depending on their sequences and the site of conjugation [24,38]. Analogs were tested in terms of their antimicrobial, anticancer, and hemolytic activities.

In terms of antimicrobial activity, previous works have shown that NST-2 (**1**) possesses strong antibacterial activity with low hemolytic effect [55,56]. The retro analog of **1**, RNST-2 (**2**), was remarkably active against both Gram-positive and Gram-negative bacteria as well as against Candida albicans with MIC values lower than the ones of **1** in all cases. This result confirmed that retro analogs possess stronger antimicrobial activity compared to parent AMPs as described in the literature [8,9,10]. Similarly, RSP-1 peptide (**3**) obtained by the addition of Lys at the C-terminal end of analog **2** also showed higher antimicrobial activities compared to NST-2 (**1**), but similar activity compared to retro NST-2 (**2**) except for *E. coli* and *P. aeruginosa* with an improved activity. This is in line with the literature, showing that a Lys addition to AMPs improves their antimicrobial activity [58,59,60]. Interestingly, RSP-4 (**7**), an analog of **2** wherein Gly^4^ was substituted with Lys, was found to be less active than NST-2 (**1**) or retro NST-2 (**2**) demonstrating that the site of Lys addition/substitution is critical in enhancing the activity of AMPs. Similarly, the site/type of conjugation of AMPs with levofloxacin influences the activity of the analogs. Conjugation with levofloxacin at the N-terminus of Phe^1^ through the β-alanine linker in analogs **6** and **8** improved activity against both Gram-positive and Gram-negative bacteria as described in the literature for some AMPs conjugated to levofloxacin. Surprisingly, and not expected due to the selective antibacterial activity of levofloxacin, conjugation to levofloxacin on Phe^1^ also improved the antifungal activity of analog **6** but not analog **8** against C. albicans. Importantly, analog **6** was also the most active analog against C. albicans with an MIC of 3.12 µM compared to 20.0 and 15.6 µM for NST-2 (**1**) and retro NST-2 (**2**), respectively. In contrast to analogs **6** and **8**, RLFP-1 (**4**) and RLFP-2 (**5**) obtained by the conjugation of levofloxacin to the side chain of Lys^14^ or Lys^3^ displayed a reduced antimicrobial effect compared to parent analog **3,** reinforcing the idea that modifications have different effects depending on the site/amino-acid modified.

Regarding anticancer activity, retro NST-2 (**2**) was to be found less active than parent NST-2 (**1**) on MCF-7 but more active on HeLa cells. In accordance with the literature, the addition of Lys at the C-terminus in analog **3** and the substitution of Gly with Lys at position 4 in analog **7** both increased the anticancer activity against MCF-7 and HeLa cells. Surprisingly, and not expected due to the selective antibacterial action of levofloxacin, the conjugation with levofloxacin at Phe^1^ (analogs **6** and **8**) resulted in a higher anticancer effect on MCF-7 cells whereas on HeLa cells, only analog **8** showed increased activity. Analogs **4** and **5** in which levofloxacin was conjugated to the side chain of Lys^14^ or Lys^3^ were found inactive with IC_50_ > 100 µM showing that the site of conjugation to levofloxacin influences the anticancer activity of the analogs as observed for antimicrobial activity.

In terms of the hemolytic effect, retro NST-2 (**2**) unfortunately was found to be more hemolytic than the parent NST-2 (**1**) (HC_50_ of 13.8 versus 90.0 µM). Regarding other analogs, when compared with their parent molecule (i.e., the analog **2**), analogs **3**, **4**, **5**, and **7**, all containing additional Lys, were found to be less hemolytic than analog **2** (HC_50_ values of 51.0, 98.9, 25.0, and >100 µM versus 13.8 µM) confirming that the addition of Lys to AMPs’ sequences reduces their hemolytic effect. Oppositely and surprisingly, as levofloxacin by itself is not hemolytic, analogs **6** and **8** in which levofloxacin was added to Phe^1^ were found to be more hemolytic than **2** (HC_50_ of 6.6 and 4.5 µM versus 13.8 µM, respectively).

Although a high percentage of alpha-helix seems to be important for the antimicrobial activity of different AMPs, our data did not find such dependency for NST-2 retro analogs. Indeed, analogs with the higher percentage of helix were not as necessary as the ones giving the lowest MIC. Although analogs **3** and **6** with 81.8 and 82.2% helix gave good activity (MIC as low as 1.56 µM), analogs **4** and **5**, with 80.1 and 80.4% helix, were the least active analogs (MIC ranging from 25 to >200 µM). Oppositely, analogs **2** and **8** with 62 and 75% helix were found very active (MIC as low as 1.56 µM). The percentage of alpha-helix in analogs does not seem to correlate with their anticancer activity. Indeed, the efficiency order in term of anticancer activity on MCF-7 was found to be **6** (82.2% helix) > **8** (75.0% helix) > **7** (56.8% helix) > **3** (81.8% helix) > **2** (62.0% helix) with **4** (80.1% helix) and **5** (80.4% helix) being inactive. Similarly, no correlation was found between the percentage of alpha-helix and hemolytic activity of the analogs, since analogs with a high percentage of alpha-helix e.g., analogs **1**, **4**, and **6** (81.5, 80.1, and 82.2% helix) have HC_50_ of 90.0, 98.9, and 6.6 µM, respectively.

As the modifications of the peptides affected both the activities and hemolytic effect of the analogs, the determination of their selectivity indexes (SIs) was needed to identify the best analog(s). Regarding antimicrobial activity, the selectivity indexes order was **3** > **1** > **2** = **7** > **6** > **4** > **8** > **5**. The retro analog of NST-2 (**2**) displayed a lower SI compared to analog **1** (i.e., 8.8 versus 14.4) showing that although this analog is more active on micro-organisms, its parallel higher hemolytic effect reduces its selectivity. The addition of Lys in analogs **3** and **7** improves their SIs by reducing their hemolytic effect compared to parent analog **2**, confirming that the addition of Lys into AMPs improves their selectivity through decreased toxicity. Regarding levofloxacin conjugate analogs, their selectivity was reduced either due to a decrease in antimicrobial effect (analogs **4** and **5**) or to an increase in their hemolytic effect (analogs **6** and **8**). Regarding anticancer activity, the selectivity indexes’ order was **1** > **7** > **3** > **4** > **6** > **2** = **8** > **5**, with all analogs being less selective than analog **1**, but when compared to analog **2** from which they originated, analogs **3**, **6**, and **7** were found to be more selective, suggesting again that the presence of additional Lys in the sequence of retro NST-2 analogs also improves their selectivity against cancer cells.

## 4. Materials and Methods

### 4.1. Reagents and Instruments

The reagents used in our experiments were 95–98% pure. Novabiochem (Hohenbrunn, Germany) and Chem-impex (Wood Dale, IL, USA) provided the Fmoc-protected amino acids, Rink amide resin, and coupling reagents. Fmoc protected Rink amide resin had a loading capacity of 0.602 mmol/g and mesh size of 200–300. HPLC grade solvents were employed. All peptides were purified with preparative HPLC (PuriFlash^®^, Interchim, Montluçon, France) by using the PFB15C18XS-250/212 column. The purity of the peptides was checked via UPLC (Agilent 1260 Infinity Diode Array, C-4 reverse-phase analytical column, 5 μm, and 150 × 4.6 mm). An electrospray ionization mass spectrometer (Q-STAR XL, Applied Biosystems, Waltham, MA, USA) with a quadrupole time-of-flight analyzer (ESI-QTOF-MS) was used for molecular mass determination. An NMR spectrometer of 600 MHz (Bruker, Fällanden, Switzerland) was used to record 1D and 2D NMR spectra. Carbon spectra were obtained by adjusting the frequency to 125 MHz.

### 4.2. Peptides Synthesis and Characterization

#### 4.2.1. Synthesis of Temporin-SHa Retro-Analogs and Their Levofloxacin Conjugates

Peptide analogs and their levofloxacin conjugates were manually synthesized by using the Fmoc solid phase method on Rink amide resin as shown in Figure 1. The resin (1 g, 0.6 mmol/g) was soaked in DMF for 2 h, followed by its treatment with 4-methylpiperidine (20%) to remove the protecting group. Fmoc-Phe-OH (6 equiv.) was then loaded on the resin with the help of 6 equivalents of *N*,*N*′-diisopropylcarbodiimide (DIC) and 2-cyano-2-(hydroxyimino) acetate (Oxyma pure) as additive to suppress racemization. After the reaction, 4-methylpiperidine in DMF (20%) was added to remove the protecting group and the next amino acid was coupled. Overall, the successive coupling was conducted by employing 3 equivalents each of Fmoc-protected amino acids, coupling reagent, and additive. Levofloxacin was coupled with peptide sequences at different positions after the removal of Alloc- and Fmoc-protecting groups. The peptides were then cleaved from the resin with TFA cocktail (94% TFA, 1% triisopropylsilane, 2.5% ethanedithiol, 2.5% water). The crude peptide was precipitated with diethyl ether and lyophilized.

#### 4.2.2. Mass and NMR Spectroscopic Analysis of Peptides

For the determination of the mass of peptides, an electrospray ionization with quadrupole time-of-flight mass spectrometry (ESI-QIT-MS) technique was used. All spectra were recorded on an Amazon Speed mass spectrometer (Bruker Daltonics GmbH, Bremen, Germany). Then, a 4500 V electrospray voltage was applied at the spraying needle with 200 °C. The sample was injected with gas flow at 15 psi. All the peptides were also characterized via ^1^H-NMR and ^13^C-NMR spectroscopy. The 600 MHz and 800 MHz NMR spectrometer (Bruker Daltonics GmbH, Bremen, Germany) was equipped with an Avance III HD console (5 RF channels, 2 receivers), with a TCI (^1^H/^13^C/^15^N/^31^P/2H) CryoProbe which is a H-optimized triple resonance NMR ‘inverse’ probe, and also a RT probe.

##### Synthesis of NST-2 Peptide (**1**)

Peptide **1** was synthesized by a solid phase peptide synthesis (SPPS) method as described earlier [65]. The overall yield was: 68.6%; ${\left[\alpha \right]}_{D}^{24}$ = −40 (c 0.22, 60% ACN, 40% H_2_O, 0.082% TFA). The purity of the peptide was confirmed with UPLC (Appendix A).

##### Synthesis of RNST-2 Peptide (**2**)

Reversing the amino acid sequence of peptide **1** via SPPS gave peptide **2**. Overall yield: 24.63%; ${\left[\alpha \right]}_{D}^{24}$ = −202 (c 0.001, MeOH). UV-Vis. (MeOH) λ_max_ (log ε): 228.0 (1.349) nm. IR (KBr, cm^−1^): 1054.60 (C-O stretching, CN stretching, NH bending), 1456.81 (C-H bending, Aromatic C=C stretching), 1636.50 (NHC=O stretching), 2927.3 (C-H stretching), 3334.86 (OH stretching) and 3853.15 (NH stretching). ^1^H NMR (d_6_-DMSO, 600 MHz): *δ*_H_ 0.72–0.81 (18H, m, (CH_3_)_2_–Leu^2, 5, 12^), 0.80–0.87 (12H, m, *δ*-CH_3_–IIe^9^, *γ*-CH_3_–IIe^9^, (CH_3_)_2_–Val^8^), 1.16 (3H, d, CH_3_–D-Ala^10^), 1.29–1.55 (18H, m, CH_2_–Leu^2, 5, 12^, *δ*-CH–Leu^2, 5, 12^), 1.30–2.39 (*β*-CH–IIe^9^, *β*-CH_2_–Met^6^, *β*-CH_2_–Lys^3^, *γ*-CH_2_–Lys^3^, *δ*-CH_2_–Lys^3^), 1.45–1.50 (1H, m, *γ*-CH–IIe^9^), 1.92 (1H, m, *β*-CH–Val^8^), 1.99 (1H, m, *δ*-CH_3_–Met^6^), 1.78–1.81 (2H, m, *γ*-CH_2_–Met^6^), 2.72 (2H, t, Δ-CH_2_–Lys^3^), 2.85–3.15 (4H, dd, CH_2_–Phe^1, 13^), 3.60 (2H, m, CH_2_–Ser^11^), 3.77–3.80 (4H, m, CH_2_–Gly^4,7^), 3.94–4.12 (6H, m, *α*-CH–Phe^1^, *α*-CH–Leu^2^, *α*-CH–Leu^5^, *α*-CH–Met^6^, *α*-CH–Leu^12^, *α*-CH–Phe^13^), 4.13–4.34 (5H, m, *α*-CH–Lys^3^, *α*-CH–Val^8^, *α*-CH–IIe^9^, *α*-CH–D-Ala^10^, *α*-CH–Ser^11^), 5.3 (1H, bs, OH–Ser^11^), 7.16–4.27 (10H, m, CH_Ar_–phe^1,13^), 7.74–7.95 (7H, m, NH–Leu^2^, NH–Gly^4^, NH–Leu^5^, NH–Met^6^, NH–Val^8^, NH–D-Ala^10^, NH–Leu^12^). 7.96–8.49 (6H, m, NH–Phe^1^, NH–Lys^3^, NH–Gly^7^, NH–IIe^9^, NH–Ser^11^, NH–Phe^13^). ^13^C-NMR (d_6_-DMSO, 150 MHz): *δ* ppm 11.17, 11.19, 14.78, 15.40, 18.36, 18.45, 19.36, 21.59, 21.64, 21.73, 21.77, 22.40, 23.10, 23.22, 23.30, 23.35, 24.23, 24.32, 24.80, 26.85, 29.69, 30.63, 31.46, 31.80, 34.63, 36.39, 37.46, 40.62, 40.85, 41.06, 42.07, 42.18, 48.40, 51.33, 52.25, 52.30,52.84, 54.08, 54.84, 54.95, 57.72, 58.03, 61.90, 126.56, 127.33, 128.35, 128.73, 129.26, 129.28, 129.39, 129.74, 137.96, 168.94, 170.11, 170.89, 170.94, 171.00, 171.48, 171.72, 171.92, 171.94, 171.99, 172.36, 172.44, 172.48, 173.36. HRMS (ESI) [M + Na]^+^ *m*/*z:* calculated for [C_68_H_111_N_15_O_14_S + Na]^+^: 1416.8053; found: 1416.8042.

##### Synthesis of RSP-1 Peptide (**3**)

Peptide **3** was synthesized by SPPS, wherein lysine was added to position-14 of retro analog **2**. Overall yield: 7.6%; ${\left[\alpha \right]}_{D}^{24}$ = −208.00 (c 0.001, MeOH). UV-Vis. (MeOH) λ_max_ (log ε): 228.0 (1.349) nm. IR (KBr, cm^−1^): 1143.30, 1201.2 (C-O stretching, CN stretching, NH bending), 1472.84 (C-H bending, Aromatic C=C stretching), 1635.72 (NHC=O stretching), 2981.04 (C-H stretching), 3356.37 (OH, NH stretching). ^1^H NMR (d_6_-DMSO, 600 MHz): *δ*_H_ 0.72–0.87 (18H, m, (CH_3_)_2_–Leu^2, 5, 12^), 0.80–0.87 (12H, m, *δ*-CH_3_–IIe^9^, *γ*-CH_3_–IIe^9^, (CH_3_)_2_–Val^8^), 1.16 (3H, d, CH_3_–D-Ala^10^), 1.29–1.55 (18H, m, CH_2_–Leu^2, 5, 12^, *δ*-CH–Leu^2, 5, 12^), 1.30–2.39 (*β*-CH–IIe^9^, *β*-CH_2_–Met^6^, *β*-CH_2_–Lys^3, 14^, *γ*-CH_2_–Lys^3, 14^, *δ*-CH_2_–Lys^3, 14^), 1.45–1.50 (1H, m, *γ*-CH–IIe^9^), 1.92 (1H, m, *β*-CH–Val^8^), 1.99 (1H, m, *δ*-CH_3_–Met^6^), 1.78–1.81 (2H, m, *γ*-CH_2_–Met^6^), 2.72 (2H, t, Δ-CH_2_–Lys^3, 14^), 2.85–3.15 (4H, dd, CH_2_–Phe^1, 13^), 3.60 (2H, m, CH_2_–Ser^11^), 3.77–3.80 (4H, m, CH_2_–Gly^4,7^), 4.04–4.20 (4H, m, *α*-CH–Phe^1^, *α*-CH–Leu^5^, *α*-CH–Val^8^, *α*-CH–IIe^9^), 4.13–4.34 (8H, m, *α*-CH–Leu^2^, *α*-CH–Lys^3^, *α*-CH–Met^6^, *α*-CH–D-Ala^10^, *α*-CH–Ser^11^, *α*-CH–Leu^12^, *α*-CH–Phe^13^, *α*-CH–Lys^14^), 5.3 (1H, bs, OH–Ser^11^), 7.16–4.27 (10H, m, CH_Ar_–phe^1,13^), 7.84–8.10 (8H, m, NH–Leu^5^, NH–Gly^7^, NH–Val^8^, NH–IIe^9^, NH–D-Ala^10^, NH–Ser^11^, NH–Leu^12^, NH–Phe^13^). 8.11–8.60 (5H, m, NH–Leu^2^, NH–Lys^3^, NH–Gly^4^, NH–Met^6^, NH–Lys^14^). ^13^C-NMR (d_6_-DMSO, 150 MHz): *δ* ppm 11.05, 15.25, 15.34, 18.12, 18.20, 19.21, 19.36, 21.44, 21.46, 21.51, 21.60, 21.70, 22.21, 22.34, 22.98, 23.00, 23.04, 23.08, 23.14, 23.19, 23.93, 24.03, 24.12, 24.53, 24.57, 26.60, 26.69, 27.15, 28.12, 29.52, 30.30, 31.47, 30.52, 31.40, 34.36, 37.10, 37.88, 38.69, 40.27, 40.83, 40.94, 41.83, 48.10, 49.17, 49.26, 51.01, 51.18, 52.54, 53.27, 57.72, 61.70, 126.29, 126.42, 127.14, 127.58, 127.81, 128.06, 128.19, 128.29, 128.52, 129.12, 129.60, 134.96, 137.65, 168.57, 169.14, 170.13, 170.62, 170.77, 170.88, 171.10, 171.51, 171.69, 171.72. HRMS (ESI) [M + Na]^+^ *m*/*z:* calculated for [C_74_H_123_N_17_O_15_S + Na]^+^: 1544.9003; found: 1544.8978.

##### Synthesis of RLFP-1 (**4**)

Peptide **4** was synthesized by SPPS, wherein lysine was added to the retro analog **2** followed by its conjugation with levofloxacin. Overall yield: 22.5%; ${\left[\alpha \right]}_{D}^{24}$ = 22.5%; ${\left[\alpha \right]}_{D}^{24}$ = +120 (c 0.001, MeOH). UV-Vis. (MeOH) λ_max_ (log ε): 206.0 (1.064) nm. IR (KBr, cm^−1^): 1196.40 (C-O stretching, CN stretching, NH bending), 1455.64 (C-H bending, Aromatic C=C stretching), 1635.96 (NHC=O stretching), 2981.07 (C-H stretching), 3335.69 (OH, NH stretching). ^1^H NMR (d_6_-DMSO, 600 MHz): *δ*_H_ 0.72–0.81 (18H, m, (CH_3_)_2_–Leu^2, 5, 12^), 0.80–0.87 (12H, m, *δ*-CH_3_–IIe^9^, *γ*-CH_3_–IIe^9^, (CH_3_)_2_–Val^8^), 1.16 (3H, d, CH_3_–D-Ala^10^), 1.29–1.55 (21H, m, 14′-CH_3_–LF, *α*-CH_2_–*b*Ala CH_2_–Leu^2, 5, 12^, *δ*-CH–Leu^2, 5, 12^), 1.30–2.39 (*β*-CH–IIe^9^, *β*-CH_2_–Met^6^, *β*-CH_2_–Lys^3,14^, *γ*-CH_2_–Lys^3,14^, *δ*-CH_2_–Lys^3,14^), 1.45–1.50 (1H, m, *γ*-CH–IIe^9^), 1.92 (1H, m, *β*-CH–Val^8^), 1.99 (1H, m, *δ*-CH_3_–Met^6^), 1.78–1.81 (2H, m, *γ*-CH_2_–Met^6^), 2.72 (6H, t, Δ-CH_2_–Lys^3,14^), 2.85–3.16 (11H, dd, 15′-CH_3_–LF, CH_2_–Phe^1, 13^, 3″ 5″-CH_2_–LF), 3.47 (2″ 6″-CH_2_–LF), 3.60 (2H, m, CH_2_–Ser^11^), 3.68–3.77 (4H, m, CH_2_–Gly^4,7^), 4.04–4.18 (5H, m, *α*-CH–Phe^1^, *α*-CH–Lys^3^, *α*-CH–Leu^5^, *α*-CH–Val^8^, *α*-CH–IIe^9^), 4.22–4.54 (11H, m, *α*-CH–Leu^2^, *α*-CH–Met^6^, *α*-CH–D-Ala^10^, *α*-CH–Ser^11^, *α*-CH–Leu^12^, *α*-CH–Phe^13^, *α*-CH–Lys^14^, 2′-CH_2_–LF), 4.84 (3′-CH–LF), 5.3 (1H, bs, OH–Ser^11^), 7.16–7.27 (10H, m, CH_Ar_–phe^1,13^), 7.53 (1H, **8′**-CH–LF), 7.75–8.00 (8H, m, NH–Lys^3^, NH–Leu^5^, NH–Met^6^, NH–Val^8^, NH–D-Ala^10^, NH–Ser^11^, NH–Leu^12^, NH–Phe^13^). 8.01–8.58 (6H, m, NH–Phe^1^, NH–Leu^2^, NH–Gly^4^, NH–Gly^7^, NH–IIe^9^, NH–Lys^14^), 8.76 (1H, 5′-CH–LF). ^13^C-NMR (d_6_-DMSO, 150 MHz): *δ* ppm 14.7, 15.4, 15.5, 18.3, 19.3, 19.3, 21.6, 21.7, 21.7, 21.8, 22.4, 22.7, 22.8, 23.1, 23.2, 23.3, 24.1, 24.2, 24.3, 24.7, 26.8, 27.3, 29.2, 29.2, 29.7, 30.6, 30.6, 31.0, 31.5, 31.7, 34.6, 36.1, 36.4, 37.1, 38.9, 40.3, 40.8, 41.0, 42.0, 42.1, 48.8, 51.3, 51.4, 52.3, 52.8, 53.3, 57.7, 58.0, 61.8, 64.4, 73.0, 129.3, 129.3, 129.4, 129.7, 133.9, 134.8, 137.7, 156.2, 162.8, 167.9, 168.9, 171.0, 171.4, 171.7, 171.8, 171.9, 172.5, 173.8. HRMS (ESI) [M + H]^+^ *m*/*z:* calculated for [C_92_H_141_FN_20_O_18_S + H]^+^: 1866.0516; found: 1866.1101.

##### Synthesis of RLFP-2 Peptide (**5**)

Peptide **5** was synthesized by SPPS, wherein lysine was added to the retro analog **2** followed by levofloxacin conjugation with lysine-3. Overall yield: 17%; ${\left[\alpha \right]}_{D}^{24}$ = +5 (c 0.001, MeOH). UV-Vis. (MeOH) λ_max_ (log ε): 206 (1.064) nm. IR (KBr, cm^−1^): 1145.49 (C-O stretching, CN stretching, NH bending), 1456.10 (C-H bending, Aromatic C=C stretching), 1627.49 (NHC=O stretching), 2981.05 (C-H stretching), 3285.26 (OH, NH stretching). ^1^H NMR (d_6_-DMSO, 600 MHz): *δ*_H_ 0.72–0.81 (18H, m, (CH_3_)_2_–Leu^2, 5, 12^), 0.80–0.87 (12H, m, *δ*-CH_3_–IIe^9^, *γ*-CH_3_–IIe^9^, (CH_3_)_2_–Val^8^), 1.16 (3H, d, CH_3_–D-Ala^10^), 1.29–1.55 (21H, m, 14′-CH_3_–LF, CH_2_–Leu^2, 5, 12^, *δ*-CH–Leu^2, 5, 12^), 1.30–2.39 (*β*-CH–IIe^9^, *β*-CH_2_–Met^6^, *β*-CH_2_–Lys^3,14^, *γ*-CH_2_–Lys^3,14^, *δ*-CH_2_–Lys^3,14^), 1.45–1.50 (1H, m, *γ*-CH–IIe^9^), 1.92 (1H, m, *β*-CH–Val^8^), 1.99 (1H, m, *δ*-CH_3_–Met^6^), 1.78–1.81 (2H, m, *γ*-CH_2_–Met^6^), 2.72 (2H, t, Δ-CH_2_–Lys^3,14^), 2.85–3.16 (11H, dd, 15′-CH_3_–LF, CH_2_–Phe^1, 13^, 3″ 5″-CH_2_–LF), 3.47 (2″ 6″-CH_2_–LF), 3.60 (2H, m, CH_2_–Ser^11^), 3.68–3.77 (4H, m, CH_2_–Gly^4,7^), 4.07–4.21 (5H, m, *α*-CH–Lys^3^, *α*-CH–Val^8^, *α*-CH–IIe^9^, *α*-CH–Leu^12^, *α*-CH–Lys^14^), 4.22–4.54 (11H, m, *α*-CH–Phe^1^, *α*-CH–Leu^2^, *α*-CH–Leu^5^, *α*-CH–Met^6^, *α*-CH–D-Ala^10^, *α*-CH–Ser^11^, *α*-CH–Phe^13^ 2′-CH_2_–LF), 4.84 (3′-CH–LF), 5.3 (1H, bs, OH–Ser^11^), 7.16–4.27 (10H, m, CH_Ar_–phe^1,13^), 7.53 (1H, **8′**-CH–LF), 7.83–8.00 (7H, m, NH–Lys^3^, NH–Leu^5^, NH–Met^6^, NH–Val^8^, NH–IIe^9^, NH–D-Ala^10^, NH–Phe^13^). 8.01–8.56 (7H, m, NH–Phe^1^, NH–Leu^2^, NH–Gly^4^, NH–Gly^7^, NH–Ser^11^, NH–Leu^12^, NH–Lys^14^), 8.76 (1H, 5′-CH–LF). ^13^C-NMR (d_6_-DMSO, 150 MHz): *δ* ppm 15.01, 15.64, 15.80, 18.09, 18.73, 19.55, 21.82, 21.88, 21.95, 22.04, 22.65, 22.74, 23.06, 23.35, 23.48, 2.54, 23.68, 24.40, 24.50, 24.61, 25.13, 26.91, 27.53, 29.52, 29.99, 30.66, 30.68, 31.45, 31.64, 31.85, 34.93, 36.45, 37.34, 37.39, 40.39, 40.54, 40.54, 40.78, 41.05, 42.56, 43.22, 47.80, 48.96, 51.78, 51.91, 52.33, 52.51, 52.87, 52.99, 53.62, 53.73, 54.04, 54.80, 55.65, 58.35, 58.78, 61.89, 64.84, 73.52, 116.77, 117.58, 118.25, 119.73, 127.03, 127.83, 128.73, 128.75, 129.11, 129.61, 130.05, 134.07, 134.88, 137.85, 141.16, 156.78, 159.07, 159.23, 159.38, 159.54, 168.29, 169.62, 169.73, 170.55, 171.16, 171.45, 171.59, 171.80, 172.11, 172.37, 172.87, 172.91, 173.20, 173.83, 174.38. LRMS (ESI) *m*/*z*: 1912.3 [M + 2Na + H]^+^

##### Synthesis of RLFP-3 (**6**)

Peptide **6** was synthesized by SPPS, wherein lysine was added to retro analog **2** followed by levofloxacin conjugation with its phenylalanine-1 via the β-alanine linker. Overall yield: 21%; ${\left[\alpha \right]}_{D}^{24}$ = −45 (c 0.001, MeOH). UV-Vis. (MeOH) λ_max_ (log ε): 224 (1.064) nm. IR (KBr, cm^−1^): 1135.15 (C-O stretching, CN stretching, NH bending), 1456.85 (C-H bending, Aromatic C=C stretching), 1670.12 (NHC=O stretching), 2972.69 (C-H stretching), 3648.95 (OH, NH stretching). ^1^H NMR (d_6_-DMSO, 600 MHz): *δ*_H_ 0.72–0.81 (18H, m, (CH_3_)_2_–Leu^2, 5, 12^), 0.80–0.87 (12H, m, *δ*-CH_3_–IIe^9^, *γ*-CH_3_–IIe^9^, (CH_3_)_2_–Val^8^), 1.16 (3H, d, CH_3_–D-Ala^10^), 1.29–1.55 (21H, m, 14′-CH_3_–LF, CH_2_–Leu^2, 5, 12^, *δ*-CH–Leu^2, 5, 12^), 1.30–2.39 (*β*-CH–IIe^9^, *β*-CH_2_–Met^6^, *β*-CH_2_–Lys^3,14^, *γ*-CH_2_–Lys^3,14^, *δ*-CH_2_–Lys^3,14^), 1.45–1.50 (1H, m, *γ*-CH–IIe^9^), 1.92 (1H, m, *β*-CH–Val^8^), 1.99 (1H, m, *δ*-CH_3_–Met^6^), 1.78–1.81 (2H, m, *γ*-CH_2_–Met^6^), 2.35 (*α*-CH_2_–*b*Ala), 2.72 (2H, t, Δ-CH_2_–Lys^3,14^), 2.85–3.16 (11H, dd, 15′-CH_3_–LF, CH_2_–Phe^1, 13^, 3″ 5″-CH_2_–LF), 3.37–3.44 (*β*-CH_2_–Ala), 3.47 (2″ 6″-CH_2_–LF), 3.60 (2H, m, CH_2_–Ser^11^), 3.66–3.80 (4H, m, CH_2_–Gly^4,7^), 4.11–4.23 (5H, m, *α*-CH–Lys^3^, *α*-CH–Val^8^, *α*-CH–IIe^9^, *α*-CH–Leu^12^, *α*-CH–Lys^14^), 4.23–4.55 (12H, m, *α*-CH–Phe^1^, *α*-CH–Leu^2^, *α*-CH–Leu^5^, *α*-CH–Met^6^, *α*-CH–D-Ala^10^, *α*-CH–Ser^11^, *α*-CH–Phe^13^, 2′-CH_2_–LF), 4.84 (3′-CH–LF), 5.3 (1H, bs, OH–Ser^11^), 7.16–7.27 (10H, m, CH_Ar_–phe^1,13^), 7.53 (1H, **8′**-CH–LF), 7.85–8.00 (8H, m, NH–Leu^2^, NH–Lys^3^, NH–Val^8^, NH–IIe^9^, NH–Ser^11^, NH–Leu^12^, NH–Phe^13^, NH–Lys^14^). 8.01–8.31 (6H, m, NH–Phe^1^, NH–Gly^4^, NH–Leu^5^, NH–Met^6^, NH–Gly^7^, NH–D-Ala^10^), 8.76 (1H, **5′**-CH–LF), 9.88 (NH–*b*Ala). ^13^C-NMR (d_6_-DMSO, 150 MHz): *δ* ppm 11.01, 14.58, 15.22, 17.9, 18.14, 18.39, 19.16, 21.40, 21.43, 21.55, 22.09, 22.18, 22.30, 22.96, 23.12, 23.29, 24.01, 24.07, 24.51, 26.61, 26.64, 29.48, 30.52, 31.30, 31.42, 31.81, 31.92, 35.29, 36.37, 37.11, 37.53, 38.70, 38.76, 40.19, 40.56, 40.81, 41.79, 41.89, 42.47, 48.01, 50.93, 51.11, 51.59, 51.98, 52.21, 52.30, 53.35, 53.75, 53.91, 53.95, 54.63, 57.27, 57.60, 61.68, 68.35, 126.28, 127.93, 128.01, 128.07, 129.13, 129.19, 137.59, 137.81, 168.47, 168.52, 169.29, 170.20, 170.57, 170.70, 170.98, 171.08, 171.29, 171.63, 171.91, 172.07, 173.20, 173.24. HRMS (ESI) [M + Na]^+^ *m*/*z:* calculated for [C_95_H_146_FN_21_O_19_S + Na + H]^+^: 1960.0785; found: 1960.0653.

##### Synthesis of RSP-4 Peptide (**7**)

Peptide **7** was prepared by SPPS, wherein glycine-4 of retro analog **2** was substituted with lysine. Overall yield: 28%; ${\left[\alpha \right]}_{D}^{24}$ = +88 (c 0.001, MeOH). UV–Vis. (MeOH) λ_max_ (log ε): 230.0 (1.979) nm. ^1^H NMR (d_6_-DMSO, 600 MHz): *δ*_H_ 0.72–0.87 (18H, m, (CH_3_)_2_–Leu^2, 5, 12^), 0.80–0.87 (12H, m, *δ*-CH_3_–IIe^9^, *γ*-CH_3_–IIe^9^, (CH_3_)_2_–Val^8^), 1.16 (3H, d, CH_3_–D-Ala^10^), 1.29–1.55 (18H, m, CH_2_–Leu^2, 5, 12^, *δ*-CH–Leu^2, 5, 12^), 1.30–2.39 (*β*-CH–IIe^9^, *β*-CH_2_–Met^6^, *β*-CH_2_–Lys^3, 4^, *γ*-CH_2_–Lys^3, 4^, *δ*-CH_2_–Lys^3, 4^), 1.45–1.50 (1H, m, *γ*-CH–IIe^9^), 1.92 (1H, m, *β*-CH–Val^8^), 1.99 (1H, m, *δ*-CH_3_–Met^6^), 1.78–1.81 (2H, m, *γ*-CH_2_–Met^6^), 2.72 (2H, t, Δ-CH_2_–Lys^3, 4^), 2.85–3.15 (4H, dd, CH_2_–Phe^1, 13^), 3.60 (2H, m, CH_2_–Ser^11^), 3.77–3.80 (4H, m, CH_2_–Gly^7^), 4.04–4.22 (5H, m, *α*-CH–Phe^1^, *α*-CH–Lys^4^, *α*-CH–Leu^5^, *α*-CH–Val^8^, *α*-CH–IIe^9^,), 4.23–4.39 (7H, m, *α*-CH–Leu^2^, *α*-CH–Lys^3^, *α*-CH–Met^6^, *α*-CH–D-Ala^10^, *α*-CH–Ser^11^, *α*-CH–Leu^12^, *α*-CH–Phe^13^), 5.3 (1H, bs, OH–Ser^11^), 7.16–4.27 (10H, m, CH_Ar_–phe^1,13^), 7.84–8.00 (7H, m, NH– Lys^4^, NH–Leu^5^, NH–Val^8^, NH–IIe^9^, NH–D-Ala^10^, NH–Leu^12^, NH–Phe^13^). 8.01–8.60 (5H, m, NH–Leu^2^, NH–Lys^3^, NH–Met^6^, NH–Gly^7^, NH–Ser^11^). ^13^C-NMR (d_6_-DMSO, 150 MHz): *δ* ppm 14.62, 15.25, 18.08, 18.13, 18.45, 19.28, 21.36, 21.45, 21.70, 22.14, 22.32, 22.94, 23.14, 24.05, 24.14, 24.55, 24.59, 26.69, 26.74, 27.17, 28.12, 29.37, 30.30, 30.58, 31.31, 31.37, 32.05, 34.37, 37.09, 37.31, 38.75, 39.07, 39.20, 39.34, 39.49, 39.62, 39.76, 39.90, 40.02, 40.48, 41.07, 41.93, 43.29, 48.16, 50.99, 51.15, 51.87, 52.04, 52.23, 52.40, 53.26, 53.86, 54.00, 57.68, 61.79, 126.27, 126.30, 126.42, 127.16, 128.11, 128.19, 128.53, 129.11, 129.16, 129.61, 134.90, 137.88, 167.97, 170.73, 171.29, 171.31, 171.51, 171.63, 171.66, 172.07. HRMS (ESI) [M + Na]^+^ *m*/*z:* calculated for [C_72_H_120_N_16_O_14_S + Na]^+^: 1487.8788; found: 1487.8763.

##### Synthesis of RLFP-4 Peptide (**8**)

Peptide **8** was synthesized by SPPS, wherein glycine-4 of retro analog **2** was substituted with lysine followed by levofloxacin conjugation at phenylalanine-1 via the β-alanine linker. Overall yield: 25%; ${\left[\alpha \right]}_{D}^{24}$ = +131.5 (c 0.001, MeOH). UV–Vis. (MeOH) λ_max_ (log ε): 225.0 (1.65) nm. ^1^H NMR (d_6_-DMSO, 600 MHz): *δ*_H_ 0.75–0.85 (18H, m, (CH_3_)_2_–Leu^2, 5, 12^), 0.78–0.83 (12H, m, *δ*-CH_3_–IIe^9^, *γ*-CH_3_–IIe^9^, (CH_3_)_2_–Val^8^), 1.16 (3H, d, CH_3_–D-Ala^10^), 1.26–1.61 (21H, m, 14′-CH_3_–LF, CH_2_–Leu^2, 5, 12^, *δ*-CH–Leu^2, 5, 12^), 1.30–2.39 (*β*-CH–IIe^9^, *β*-CH_2_–Met^6^, *β*-CH_2_–Lys^3,4^, *γ*-CH_2_–Lys^3,4^, *δ*-CH_2_–Lys^3,4^), 1.45–1.50 (1H, m, *γ*-CH–IIe^9^), 1.92 (1H, m, *β*-CH–Val^8^), 2.00 (1H, m, *δ*-CH_3_–Met^6^), 1.78–1.81 (2H, m, *γ*-CH_2_–Met^6^), 2.35 (*α*-CH_2_–*b*Ala), 2.72 (2H, t, Δ-CH_2_–Lys^3,4^), 2.85–3.16 (11H, dd, 15′-CH_3_–LF, CH_2_–Phe^1, 13^, 3′ 5′-CH_2_–LF), 3.37–3.44 (*β*-CH_2_–*b*Ala), 3.47 (2′ 6′-CH_2_–LF), 3.61–3.85 (2H, m, CH_2_–Ser^11^), 3.66–3.84 (4H, m, CH_2_–Gly^7^), 4.09–4.22 (4H, m, *α*-CH–Lys^4^, *α*-CH–Leu^5^, *α*-CH–IIe^9^, *α*-CH–Leu^12^,), 4.23–4.54 (8H, m, *α*-CH–Phe^1^, *α*-CH–Leu^2^, *α*-CH–Lys^3^, *α*-CH–Met^6^, *α*-CH–Val^8^, *α*-CH–D-Ala^10^, *α*-CH–Ser^11^, *α*-CH–Phe^13^, 2′ 6′-CH_2_–LF), 4.84 (3′-CH–LF), 5.3 (1H, bs, OH–Ser^11^), 7.16–7.27 (10H, m, CH_Ar_–phe^1,13^), 7.53 (1H, **8′**-CH–LF), 7.80–8.03 (8H, m, NH–Lys^3^, NH–Lys^4^, NH–Met^6^, NH–Val^8^, NH–IIe^9^, NH–D-Ala^10^, NH–Ser^11^, NH–Leu^12^). 8.04–9.88 (6H, m, NH–Phe^1^, NH–Leu^2^, NH–Leu^5^, NH–Gly^7^, NH–Phe^13^), 8.76 (1H, **5′**-CH–LF), 9.88 (NH–*b*Ala). ^13^C-NMR (d_6_-DMSO, 150 MHz): *δ* ppm 10.98, 14.57, 15.19, 17.89, 18.05, 18.39, 19.15, 21.38, 21.41, 21.59, 22.11, 22.15, 22.17, 22.21, 22.89, 23.05, 23.09, 23.26, 23.98, 24.06, 24.08, 24.50, 26.62, 26.64, 29.30, 30.55, 31.26, 32.03, 34.92, 35.30, 36.30, 37.21, 37.29, 40.42, 40.60, 41.81, 42.45, 47.30, 47.97, 50.92, 51.13, 51.90, 52.14, 52.19, 5333, 53.71, 53.92, 54.40, 57.26, 57.52, 61.74, 68.32, 124.22, 126.09, 126.20, 127.92, 128.02, 129.02, 129.06, 137.77, 137.82, 145.01, 163.90, 168.46, 170.45, 170.53, 170.99, 171.18, 171.20, 171.25, 171.51, 171.92, 171.97, 172.88, 173.93, 173.94. HRMS (ESI) [M + Na]^+^ *m*/*z:* calculated for [C_93_H_143_FN_20_O_18_S + Na + H]^+^: 1903.0570; found: 1903.0484.

#### 4.2.3. Circular Dichroism (CD) and Secondary Structure Analysis

The far-ultraviolet circular dichroism (CD) spectra were recorded using a JASCO J-810 spectropolarimeter (Jasco, Tokyo, Japan), with measurements taken in a quartz cuvette with a 10 mm path length. The temperature was kept at 22 °C, and the instrument was calibrated with D-(+)-10-camphorsulfonic acid. Peptides were dissolved in 20 mM SDS to a final concentration of 15 µM. The CD spectra were collected over a wavelength range of 190 to 260 nm with a bandwidth of 2 nm. Each spectrum was recorded with ten consecutive scans at a rate of 50 nm/min, and the baseline was obtained under identical conditions. Secondary structures, i.e., the percentage of alpha-helix and other secondary structures, were quantified from CD data using Bestsel software (https://bestsel.elte.hu/index.php, accessed on 22 October 2024).

### 4.3. Biological Studies

#### 4.3.1. Antibacterial Assay

The antibacterial activity of peptide analogs was tested against different bacterial strains, i.e., *Salmonella enterica* subsp. *enterica* serotype Typhimurium [68] or *Salmonella typhi* (ATCC 14028), *Staphylococcus aureus* (NCTC 13277), *Bacillus subtilis* (ATCC 23857), *Escherichia coli* (ATCC 25922), and *Pseudomonas aeruginosa* (ATCC 10145). These strains were obtained from the microbial bank of Dr. Panjwani Center for Molecular Medicine and Drug research (PCMD), International Center for Chemical and Biological Sciences (ICCBS), University of Karachi. Colonies of these bacterial strains were grown in respective agar media, then inoculated in Mueller Hinton (MH) broth (Oxoid, UK) and incubated at 37 °C overnight. The peptide solutions were first diluted 1 in 100 (from the stock solution at 20 mM) and then further two-fold dilution in MH broth in a sterile 96-well plate was performed, resulting in 100 µL of broth containing increasing concentrations of analogs. Bacteria in their exponential growth phase were diluted in MH broth, and 100 µL of this suspension was added in each well. This resulted in a final 200 µL suspension containing 0.5−1.0 × 10^6^ CFU/mL. Then, these plates were incubated at 37 °C for 20–22 h. MIC (Minimum Inhibitory Concentration) values were determined as the concentrations of peptides causing >99% inhibition of bacterial growth.

#### 4.3.2. Antifungal Assay

*Candida albicans* (ATCC 36082) was cultured on Sabourad Dextrose Agar (SDA). A single colony was inoculated in Sabouraud Dextrose Broth (SDB) medium and grown overnight at 37 °C while shaking. The next day, the susceptibility of *C. albicans* was tested using a broth microdilution assay. Peptide analogs were diluted in SDB media by two-fold dilution, resulting in 100 µL of medium containing increasing concentrations of analogs. The overnight culture (turbidity at OD_600_ = 1) was 1000× diluted in SDB. Then 100 µL of this suspension was added to each well, resulting in a final 200 µL suspension containing 2–4 × 10^5^ CFU/mL. The plate was incubated at 37 °C for 24 h. The next day, MIC (Minimum Inhibitory Concentration) values were determined as the concentrations of peptides causing > 99% inhibition of yeast growth.

#### 4.3.3. Antiproliferative Assay

Human breast cancer (MCF-7) and human cervical cancer (HeLa) cells were obtained from the cell culture bank of PCMD in ICCBS, University of Karachi. These cells were maintained in Dulbecco’s Modified Eagle Medium (DMEM), supplemented with 10% fetal bovine serum, and incubated at 37 °C and 5% CO_2_. The antiproliferative activity of the samples was determined by a MTT assay. Briefly, the cells were washed with PBS and trypsinized. After determination of the cell number using Malassez cell counting, 100 µL of the cell suspension were seeded in a 96-well plate at the density of 6000 cells/well and incubated at 37 °C and 5% CO_2_. After 24 h, the cells were treated with increasing concentrations of analogs. The next day, media was removed and 200 µL of MTT dye (0.5 mg/mL) was added in each well and incubated for 3 h at 37 °C in 5% CO_2_ incubator. Then, the media was removed and 100 µL of DMSO was added to solubilize formazan crystals. After one minute of shaking, the absorbance was recorded at 540 nm in microplate reader (Multiskan GO, ThermoScientific, Waltham, MA, USA). Finally, the percent inhibition of proliferation was calculated using the following formula:$$\%\;inhibition\;of\;proliferation=\frac{O.D\;of\;treated\;well-O.D\;of\;media\;control}{O.D\;of\;untreated\;control-O.D\;of\;media\;control}\times 100$$

#### 4.3.4. Hemolytic Assay

For the hemolytic assay, fresh human blood (2 mL) was obtained from a healthy donor in EDTA tube while following the protocol of approval from the Independent Ethics Committee of ICCBS (approval number ICCBS/IEC-047-HB-2019/Protocol/1.0). The blood was centrifuged and upper supernatant plasma was removed. The cell pellets were washed with sterile PBS three times. The washed blood cells were then diluted 25 times to make 4% concentration of the initial blood cells. Red blood cells (500 µL) were treated with increasing concentrations of analogs, with triton X-100 at 0.1% being used as positive control giving 100% hemolysis. After 1 h of incubation at 37 °C, tubes were centrifuged at 800 rpm for 5 min to pellet down red blood cells. After centrifugation, 200 µL of supernatant were transferred into a 96 well plate and the absorbance was recorded at 576 nm using a microplate reader (MultiSkan Go, ThermoScientific). Then, the following formula was applied to calculate percent hemolysis:$$\%\;Hemolysis=\frac{O.D\;of\;test\;sample-O.D\;of\;PBS\;control}{O.D\;of\;Tritox\times positive\;control-O.D\;of\;PBS\;control}\times 100$$

#### 4.3.5. Atomic Force Microscopy (AFM) Imaging

Different bacteria (*S. typhi*, *E. coli*, and *P. aeruginosa*) adjusted to 2–3 × 10^7^ CFU/mL were treated with 2XMIC values of analogs and incubated overnight at 37 °C. The next day, the bacteria were washed, dispersed in sterile pure grade water, dispensed in poly-L-lysine-coated mica slides, and left for air drying. Changes in the morphology of the bacteria were studied with an atomic force microscope (Agilent 5500, Chandler, AZ, USA). The whole analysis was carried out in tapping mode. Images were collected and optimized at a scan velocity of 1–5 µm/s and 512 × 512-line resolution and processed through PicoView 1.2 imaging software.

## 5. Conclusions

In this study, we explored the efficacy enhancement of an antimicrobial peptide, NST-2 (**1**), through retro analog synthesis, lysine addition/substitution, and levofloxacin conjugation. Although analog **2** corresponding to the retro analog of NST-2 was found to be more active in bacteria and fungi, its higher hemolytic effect compared to parent NST-2 resulted in a reduction in its selectivity index and therapeutic potential. The addition of lysine (analog **3**) and lysine substitution (analog **7**) reduced the hemolytic effect of the analogs resulting in safer peptides. Conjugation with levofloxacin on the lysine side chain (in analogs **4** and **5**) was found to decrease the hemolytic effect of the analogs but unfortunately also their antimicrobial and anticancer activities. Finally, conjugates obtained through the addition of levofloxacin at the N-terminus of the peptide via the β-alanine linker (analogs **6** and **8**) possessed increased antimicrobial and anticancer activities but also unfortunately an increased hemolytic effect, resulting in less safe/selective analogs. In conclusion, lysine addition/substitution and levofloxacin conjugation to the retro analog of NST-2, at least at the N-terminal position through the β-alanine linker, were found to enhance their antimicrobial/anticancer activity and/or to decrease their hemolytic effect, enhancing their therapeutic potential.

## Data Availability

All data are given in the main manuscript and Appendix A.

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
