# Peer review of "Synthesis of Temporin-SHa Retro Analogs with Lysine Addition/Substitution and Antibiotic Conjugation to Enhance Antibacterial, Antifungal, and Anticancer Activities"

_antibiotics, 2024, doi:10.3390/antibiotics13121213_

Round 1
Reviewer 1 Report
Comments and Suggestions for Authors
This manuscript reports design, chemical synthesis and activity studies of analogs from frog derived AMP temporin SH. AMPs are naturally occurring host defence molecules which have high potential for applications in infectious diseases. Therefore, structure, activity correlation studies of AMPs are informative an d essential. Temporins are a large group of short AMPs initially isolated from European frog skins. Founding member of temporins are temporin A, temporin B and temporin L. Temporins show selective activity for gram-positive bacteria and less activity for gram-negative bacteria. LPS outer membrane induced aggregation of temporins limiting their antibacterial activity (Biochim Biophys Acta. 2009, 788(8):1610-9. doi: 10.1016/j.bbamem.2009.04.021, J Biol Chem. 2008 ;283(34):22907-17. doi: 10.1074/jbc.M800495200., J Biol Chem. 2011;286(27):24394-406. doi: 10.1074/jbc.M110.189662). These key references should be included in the manuscript. The current study systematically evaluated several analogs temporin SHa. The starting peptide used in the study is a previously investigated one termed NST2 and based on it, seven analogs were prepared and tested for antibacterial, antifungal, anticancer and hemolytic properties. Also, effects of these analogs were examined by AFM in G- bacteria. Experiments were carefully done, and conclusions made are valid. I have some minor comments that may be addressed.
1. Introduction of cationic Lys in the sequence can enhance antibacterial activity of temporins. A study demonstrated superior activity of temporins against Gram-negative bacteria by incorporating LPS binding peptide motif (Antimicrob Agents Chemother. 2014;58(4):1987-96. doi: 10.1128/AAC.02321-13). Also, lys inclusion in temporin L augment aggregation and activity (Antimicrob Agents Chemother. 2013 57(6):2457-66. doi: 10.1128/AAC.00169-13). Authors should include these works to improve the “Discussion” section, which is currently reviewing the results.
2. Table 3 shows estimated population of secondary structures of the peptides. What does antiparallel referring to? Do these peptides form b-sheet in SDS?
3. MIC values against Gram-negative strains are generally high. Authors should provide some explanations, based on previous reports on temporins.
4. The AFM images are of low resolution, quality of the images should be improved.
5. Why did authors obtain images for Gram-negative bacteria?
Author Response
Reviewer 1 Comments (Our responses in blue color)
We thanks Reviewer 1 for his/her constructive comments that helped us to improve our manuscript.
Regards
This manuscript reports design, chemical synthesis and activity studies of analogs from frog derived AMP temporin SH. AMPs are naturally occurring host defence molecules which have high potential for applications in infectious diseases. Therefore, structure, activity correlation studies of AMPs are informative and essential. Temporins are a large group of short AMPs initially isolated from European frog skins. Founding member of temporins are temporin A, temporin B and temporin L. Temporins show selective activity for gram-positive bacteria and less activity for gram-negative bacteria. LPS outer membrane induced aggregation of temporins limiting their antibacterial activity (Biochim Biophys Acta. 2009, 788(8):1610-9. doi: 10.1016/j.bbamem.2009.04.021, J Biol Chem. 2008 ;283(34):22907-17. doi: 10.1074/jbc.M800495200., J Biol Chem. 2011;286(27):24394-406. doi: 10.1074/jbc.M110.189662). These key references should be included in the manuscript.
Following paragraph was added in the introduction section (on page 2) with the following key references:
Temporins are a diverse group of short AMPs, which were originally isolated from the skin of European frogs [41-43]. The founding members of this group of AMPs include temporin A, temporin B, and temporin L [44, 45]. Temporins exhibit selective antibacterial activity against Gram-positive bacteria, with significantly lower efficacy against Gram-negative bacteria. Their reduced activity against Gram-negative strains is attributed to lipopolysaccharide (LPS)-induced aggregation in the outer membrane, which limits their antibacterial effectiveness [46-51].
- D’Andrea L.D, and Romanelli A. Temporins: multifunctional peptides from frog skin. International Journal of Molecular Sciences 2023. 24(6): p. 5426.
- Simmaco, M., Mignogna, G., Canofeni, S., Miele, R., Mangoni, M. L., and Barra, D. Temporins, antimicrobial peptides from the European red frog Rana temporaria. European Journal of Biochemistry,1996. 242(3): p. 788-792.
- Mangoni, M.L. Temporins, anti-infective peptides with expanding properties. Cellular and Molecular Life Sciences CMLS, 2006. 63: p. 1060-1069.
- Mahalka, A.K. and Kinnunen, P. K. Binding of amphipathic α-helical antimicrobial peptides to lipid membranes: Lessons from temporins B and L. Biochimica et Biophysica Acta (BBA)-Biomembranes, 2009. 1788(8): p. 1600-1609.
- Romero, S.M., Cardillo, A. B., Martínez Ceron, M. C., Camperi, S. A., & Giudicessi, S. L.Temporins: an approach of potential pharmaceutic candidates. Surgical Infections, 2020. 21(4): p. 309-322.
- Mangoni, M.L. and Shai, Y. Temporins and their synergism against Gram-negative bacteria and in lipopolysaccharide detoxification. Biochimica et Biophysica Acta (BBA)-Biomembranes, 2009. 1788(8): p. 1610-1619.
- Mangoni, M.L., and Shai, Y. Lipopolysaccharide, a key molecule involved in the synergism between temporins in inhibiting bacterial growth and in endotoxin neutralization. Biochimica et Biophysica Acta (BBA)-Biomembranes, 2008. 283(34): p. 22907-22917.
- Bhunia, A., Saravanan, R., Mohanram, H., Mangoni, M. L., and Bhattacharjya, S. NMR structures and interactions of temporin-1Tl and temporin-1Tb with lipopolysaccharide micelles: mechanistic insights into outer membrane permeabilization and synergistic activity. Journal of Biological Chemistry, 2011. 286(27): p. 24394-24406.
- Bhattacharjya, S. NMR structures and interactions of antimicrobial peptides with lipopolysaccharide: connecting structures to functions. Current Topics in Medicinal Chemistry, 2016. 16(1): p. 4-15.
- Pulido, D., Nogués, M., Boix, E., and Torrent, M. Lipopolysaccharide neutralization by antimicrobial peptides: a gambit in the innate host defense strategy. Journal of Innate Immunity, 2012. 4(4): p. 327-336.
- Schmidtchen, A., and Malmsten, M. (Lipo) polysaccharide interactions of antimicrobial peptides. Journal of colloid and interface science,2015. 449: p. 136-142.
The current study systematically evaluated several analogs temporin SHa. The starting peptide used in the study is a previously investigated one termed NST2 and based on it, seven analogs were prepared and tested for antibacterial, antifungal, anticancer and hemolytic properties. Also, effects of these analogs were examined by AFM in G- bacteria. Experiments were carefully done, and conclusions made are valid. I have some minor comments that may be addressed.
Thank you for the positive feedback and thoughtful comments. We appreciate your recognition of our study’s systematic approach and valid conclusion. We will carefully address the minor comments to further enhance the manuscript.
- Introduction of cationic Lys in the sequence can enhance antibacterial activity of temporins. A study demonstrated superior activity of temporins against Gram-negative bacteria by incorporating LPS binding peptide motif (Antimicrob Agents Chemother. 2014;58(4):1987-96. doi: 10.1128/AAC.02321-13). Also, lys inclusion in temporin L augment aggregation and activity (Antimicrob Agents Chemother. 2013 57(6):2457-66. doi: 10.1128/AAC.00169-13). Authors should include these works to improve the “Discussion” section, which is currently reviewing the results.
Following paragraph was added in the discussion section on page 7:
Incorporation of cationic lysine residue reduces the aggregation process in aqueous environment, thereby enhancing the oligomerization as it binds to lipopolysaccharides (LPS). This interaction results in the disruption of LPS aggregates, and helps to neutralize the LPS-induced inflammation in animal models, including mice and rats [66]. Furthermore, the conjugation of an LPS-binding motif to an antimicrobial peptide (AMP) can effectively neutralize endotoxins and disrupt the bacterial outer cell wall. This motif, referred to as the “boomerang motif”, represents a promising strategy for designing cell wall permeabilizing peptides [67].
- Srivastava, S., and Ghosh, J. K. Introduction of a lysine residue promotes aggregation of temporin L in lipopolysaccharides and augmentation of its antiendotoxin property. Antimicrobial Agents and Chemotherapy, 2013. 57(6): p. 2457-2466.
- Mohanram, H., and Bhattacharjya, S. Resurrecting inactive antimicrobial peptides from the lipopolysaccharide trap. Antimicrobial agents and chemotherapy, 2014. 58(4): p. 1987-19
Table 3 shows estimated population of secondary structures of the peptides. What does antiparallel referring to? Do these peptides form b-sheet in SDS?
The BeStSel (Beta Structure Selection) software was employed to estimate population of secondary structures of the peptides. The software is specifically designed to predict secondary structures, including parallel and antiparallel β-sheets, from CD spectra as demonstrated in multiple studies and its documentation. It achieves this by decomposing spectra into eight components, including α-helices, β-sheets (parallel and antiparallel with three degrees of twist), turns, and others, based on the Dictionary of Secondary Structure of Proteins (DSSP). Therefore, BeStSel reliably predicts parallel and antiparallel β-sheets, as supported by its validated computational framework and many published studies. For more details, kindly see the attached document, as well as a recent paper appeared in Nucleic Acids Res. 50 (2022) w90 (https://doi.org/10.1093/nar/gkac345). However, for the sake of clarity, following paragraph was added in the beginning of sub-section 2.1:
Sodium dodecyl sulfate (SDS) is commonly used to denature proteins as it disrupts non-covalent interactions. Peptides can form β-sheets depending on their hydrophobic residues, resistance to SDS-induced denaturation, helical content, and interactions with SDS micelles [61, 62]. Antiparallel refers to one of the orientations of β-sheets, wherein the strands align in opposite directions. This arrangement, along with parallel and twisted configurations, adds to the variety of structural patterns that can be observed in spectral analyses. For precise and detailed structural information derived from CD spectra, the Be-ta Structure Selection (BeStSel) method has become a widely utilized tool nowadays [63, 64].
- Parker, W. and Song, P. S., Protein structures in SDS micelle-protein complexes. Biophysical Journal,1992. 61(5): p. 1435-1439.
- Rozek, A., C.L. Friedrich, and R.E. Hancock, Structure of the bovine antimicrobial peptide indolicidin bound to dodecylphosphocholine and sodium dodecyl sulfate micelles. Biochemistry,2000. 39(51): p. 15765-15774.
- Micsonai, A., Moussong, E., Wien, F., Boros, E., Vadászi, H., Murvai, N., Lee, Y. H., Molnar, T., Refregiers, M., Goto, Y., Tantos, A., and Kardos, J. BeStSel: webserver for secondary structure and fold prediction for protein CD spectroscopy. 2022. Nucleic Acids Research, 50(W1): p. W90-W98.
- Micsonai, A., Moussong, É., Wien, F., Boros, E., Vadászi, H., Murvai, N., Lee, Y. H., Molnar, T., Refregiers, M., Goto, Y., Tantos, A., and Kardos, J. Bestsel: Updated webserver for secondary structure and fold prediction for protein CD spectroscopy. Biophysical Journal, 2023. 122(3): p. 179a.
MIC values against Gram-negative strains are generally high. Authors should provide some explanations, based on previous reports on temporins.
Following paragraph was added in the introduction section (page 2):
Temporin-SHa has been well-documented for its strong antibacterial activity against Gram-positive bacteria, including Staphylococcus aureus, Bacillus subtilis, and Enterococcus faecalis, while exhibiting comparatively lower activity against Gram-negative bacteria, such as Escherichia coli, Pseudomonas aeruginosa, Acinetobacter baumannii, and Klebsiella pneumoniae. Similarly, D-analog substitutions of temporin-SHa have also demonstrated potent activity against Gram-positive bacteria but reduced efficacy against Gram-negative strains. This lower activity against Gram-negative bacteria is likely due to differences in membrane composition and permeability barriers, which are characteristic of Gram-negative pathogens. Consequently, the analogs presented in this manuscript also exhibit high MIC values against Gram-negative bacteria compared to Gram-positive counterparts [52-54].
- Khan AI, Nazir S, Haque MNU, Maharjan R, Khan FA, Olleik H, Courvoisier-Dezord E, Maresca M, and Shaheen F. Synthesis of Second-Generation Analogs of Temporin-SHa Peptide Having Broad-Spectrum Antibacterial and Anticancer Effects. Antibiotics 2024. 13(8): p. 758.
- Khan, A.I.; Nazir, S.; Ullah, A.; Haque, M.N.u.; Maharjan, R.; Simjee, S.U.; Olleik, H.; Dezord, E.C.; Maresca, M.; and Shaheen, F. Design, synthesis and characterization of [G10a]-Temporin SHa dendrimers as dual inhibitors of cancer and pathogenic microbes. Biomolecules 2022. 12(6): p. 770.
- Raja Z, Andre S, Abbasi F, Humblot V, Lequin O, Bouceba T, Correia I, Casale S, Foulon T, Sereno D, Oury B, and Ladram A. Insight into the mechanism of action of temporin-SHa, a new broad-spectrum antiparasitic and antibacterial agent. PLoS One 2017. 12(3): p. e0174024
The AFM images are of low resolution, quality of the images should be improved.
Original high resolution AFM images have been provided now.
Why did authors obtain images for Gram-negative bacteria?
As highlighted in response to comment no. 3, the original peptides (temporin-SHa, [G10a]-SHa, and [G4a]-SHa) exhibited limited activity against Gram-negative bacteria. In the present study, the newly synthesized analogs demonstrated improved activity against Gram-negative strains (E. coli, S. typhi, and P. aeruginosa). AFM images were obtained for these bacteria to highlight the enhanced activity and emphasize the new strategy employed to improve efficacy against Gram-negative bacteria.

Reviewer 2 Report
Comments and Suggestions for Authors
The manuscript describes a systematic approach to evaluating a peptide-small molecule conjugate with improved activity and toxicity.
1. In the introduction, the authors do a nice job of introducing the approaches many groups have used to improve AMP activity. However, they did not include the use of non-canonical amino acids OUTSIDE of D-amino acids. Many groups have used these for detailed SAR understanding and application, so they should be included.
2. Similarly, the authors ignore the development of peptidomimetics, which have gained great interest in the field. This should also be included in the introduction.
3. The authors use the abbreviation S. typhi in the manuscript. While this is common nomenclature, the authors should include the whole name in the methods section at a minimum (Salmonella enterica subsp. enterica serotype Typhimurium); https://pmc.ncbi.nlm.nih.gov/articles/PMC86943/
4. For clarity, the authors should consider moving the NMR assignments, currently in the methods section, to the supplement.
5. There are multiple instances throughout the manuscript where bacterial names are not italicized (e.g. lines 479-480; figure legend 2, etc)
6. the resolution of figures 2-4 is not high enough. The axes are not legible.
7. Table 5 should be reformatted. the HeLA and MCF-7 cells do not indicate the values are IC50, nor are there units in the table. This is in contrast to the hemolysis column. Importantly, the columns are not equivalently labeled- the first two have cell type while the third has assay type. The hemolysis column should be changed to "red blood cells"
8. While i understand the intent of table 4, the authors should include a full tabular analysis in the supplement - EACH microorganism compared to EACH cell type for SI.
9. In table 4, the authors should indicate in the table which microorganism and which cell type generates the SI. This is important since they will not be the same throughout the table.
10. The authors should include MICs for levofloxacin alone. These may be available in the literature, but are a necessary comparison for the data.
11. the authors do not discuss mechanism of action. Is the hypothesis that the peptide is still disrupting the membrane? If that is the main active form, what roles does the LF play, and vice versa? While i understand the mechanistic studies are beyond the scope of this paper, the authors should propose some mechanisms for future investigation.
12. overall the manuscript could use a proofread for tense/pluralization.
Author Response
We thanks Reviewer 2 for his/her constructive comments that helped us to improve our manuscript.
Regards
Reviewer 2 Comments (Our responses in blue color)
The manuscript describes a systematic approach to evaluating a peptide-small molecule conjugate with improved activity and toxicity.
- In the introduction, the authors do a nice job of introducing the approaches many groups have used to improve AMP activity. However, they did not include the use of non-canonical amino acids OUTSIDE of D-amino acids. Many groups have used these for detailed SAR understanding and application, so they should be included.
Following paragraph was added on page 2 of the introduction section:
The incorporation of non-canonical amino acids can significantly enhance the stability of peptide and improve their biological functions [11, 12]. These amino acids have also been utilized as secondary metabolites, thus performing the physiological processes. Although their role is not fully understood yet, they hold great potential for optimizing the overall activity of peptides and molecules. Notable examples include seralasin for hypertension, Icatibant for enhance stability, and carbetocin for postpartum hemorrhage [13, 14].
- Du, Y., Li, L., Zheng, Y., Liu, J., Gong, J., Qiu, Z., Li, Y., Qiao, J., and Huo, Y. X. Incorporation of non-canonical amino acids into antimicrobial peptides: Advances, Challenges, and Perspectives. Applied and Environmental Microbiology, 2022. 88(23): p. e01617-22.
- Baumann, T., Nickling, J. H., Bartholomae, M., Buivydas, A., Kuipers, O. P., and Budisa, N. Prospects of in vivo incorporation of non-canonical amino acids for the chemical diversification of antimicrobial peptides. Frontiers in Microbiology, 2017. 8: p. 124.
- Castro, T.G., Melle-Franco, M., Sousa, C. E., Cavaco-Paulo, A., and Marcos, J. C. Non-Canonical Amino Acids as Building Blocks for Peptidomimetics: Structure, Function, and Applications. Biomolecules, 2023. 13(6): p. 981.
- Zou, H., Li, L., Zhang, T., Shi, M., Zhang, N., Huang, J., and Xian, M. Biosynthesis and biotechnological application of non-canonical amino acids: Complex and unclear. Biotechnology Advances,2018. 36(7): p. 1917-1927.
Similarly, the authors ignore the development of peptidomimetics, which have gained great interest in the field. This should also be included in the introduction.
Following paragraph was added on page 2 of the introduction section:
Peptide analogs, known as peptidomimetics, are designed to mimic the structural elements of natural peptides to enhance biological functions and mitigate the drawbacks of natural peptides [15-17]. Notable examples include triazolyl peptidomimetics, which have shown promise as enzyme inhibitors as well as anticancer, antibacterial, and anti-fungal agents [18, 19].
- Wu, Y.-D. and Gellman, S., Peptidomimetics. Accounts of Chemical Research,2008, p. 1231-1232.
- Lenci, E. and Trabocchi, A. Peptidomimetic toolbox for drug discovery. Chemical Society Reviews, 2020. 49(11): p. 3262-3277.
- Vagner, J., H. Qu, and Hruby, V. J. Peptidomimetics, a synthetic tool of drug discovery. Current Opinion in Chemical Biology, 2008. 12(3): p. 292-296.
- Del Gatto, A., Cobb, S. L., Zhang, J., and Zaccaro, L. Peptidomimetics: Synthetic tools for drug discovery and development. Frontiers in Chemistry, 2021, 9,. p. 802120.
The authors use the abbreviation S. typhi in the manuscript. While this is common nomenclature, the authors should include the whole name in the methods section at a minimum (Salmonella entericasubsp. enterica serotype Typhimurium); https://pmc.ncbi.nlm.nih.gov/articles/PMC86943/
In the methodology section, we have replaced Salmonella typhi with
Salmonella enterica subsp. enterica serovar Typhimurium str. ATCC 14028 [68]. In table and result, we have reported the short abbreviation as S. typhi.
- Brenner, F. W., Villar, R. G., Angulo, F. J., Tauxe, R., and Swaminathan, B. Salmonella nomenclature. Journal of Clinical Microbiology, 2000. 38(7): p. 2465-2467.
For clarity, the authors should consider moving the NMR assignments, currently in the methods section, to the supplement.
We appreciate the suggestion; however, the NMR data is integral to the synthesis and characterization of each peptide. It complements UV, FTIR, and mass spectrometry in confirming structural integrity and enhancing reproducibility. Moving it to the supplementary section would disconnect it from the synthesis methods, reducing clarity. Including it in the methods section ensures a complete and coherent presentation of our peptide preparation process.
There are multiple instances throughout the manuscript where bacterial names are not italicized (e.g. lines 479-480; figure legend 2, etc)
All the bacteria names in lines 479-480; figure legend 2 and elsewhere in the manuscript were now italicized.
the resolution of figures 2-4 is not high enough. The axes are not legible.
Original high resolution AFM images have been provided now.
Table 5 should be reformatted. the HeLA and MCF-7 cells do not indicate the values are IC50, nor are there units in the table. This is in contrast to the hemolysis column. Importantly, the columns are not equivalently labeled- the first two have cell type while the third has assay type. The hemolysis column should be changed to "red blood cells"
In Table 5, all the results were expressed in µM. For the cancer cell lines (HeLa and MCF-7), the IC50 values represented the concentration (µM) at which 50% inhibition was observed. Similarly, for hemolysis, the HC50 values denoted the concentration (µM) required to induce 50% hemolysis.
While i understand the intent of table 4, the authors should include a full tabular analysis in the supplement - EACH microorganism compared to EACH cell type for SI.
We prefer to maintain the Table 4 in the main text and not as supplementary materials. But we will add informations as suggested in point 9.
In table 4, the authors should indicate in the table which microorganism and which cell type generates the SI. This is important since they will not be the same throughout the table.
The information is now provided in the revised version of the manuscript.
The authors should include MICs for levofloxacin alone. These may be available in the literature, but are a necessary comparison for the data.
MIC values of levofloxacin have now been added in Table 4.
The authors do not discuss mechanism of action. Is the hypothesis that the peptide is still disrupting the membrane? If that is the main active form, what roles does the LF play, and vice versa? While i understand the mechanistic studies are beyond the scope of this paper, the authors should propose some mechanisms for future investigation.
The provided data demonstrate that the peptides cause bacterial lysis through membrane disruption (as does temporin SHa). The addition of Levo does not modify their mechanism of action based on obtained data.
Overall the manuscript could use a proofread for tense/pluralization.
Thank you for the suggestion. We have carefully proofread the manuscript to address issues with tense and pluralization to ensure clarity and grammatical accuracy.

Reviewer 3 Report
Comments and Suggestions for Authors
In this manuscript, the author reports studies on Temporin-SHa Retro Analogs, including its retro-analog NST-2, NST-2 with additional/substituted lysine, and their drug conjugates. The author evaluates these analogs' antibacterial, antifungal, anti-cancer, and hemolytic properties, demonstrating their potential as AMPs.Comments:
- Figure 1 is a well-designed figure showing all the analogs and their stepwise design. However, the author should remove the arrows between #4, 5, 6, 7, and 8, as these are derived from the last sample. Additionally, most of the information in this scheme is repetitive, as it is also shown in Table 1. Please consider rearranging for better clarity.
- For the CD results, how reliable are the structure quantifications with drug conjugates? Were the optical properties of the drug taken into consideration?
- Figures 2-4: The image quality appears to be very low on my screen. Additionally, from the captions, it seems that the three experiments used different groups of AMP analogs. Please explain the rationale for this in the context of the study.
- The manuscript would benefit from control experiments, if not already reported, on levofloxacin's antibacterial, anti-cancer, and hemolytic properties.
Author Response
Dear Reviewer,
We would like to thank you for your valuable comments and suggestions that help us improving our manuscript. Please find below our answers.
Regards
Dr M Maresca
Reviewer 3 Comments (Our responses in blue color)
In this manuscript, the author reports studies on Temporin-SHa Retro Analogs, including its retro-analog NST-2, NST-2 with additional/substituted lysine, and their drug conjugates. The author evaluates these analogs' antibacterial, antifungal, anti-cancer, and hemolytic properties, demonstrating their potential as AMPs. Comments:
- Figure 1 is a well-designed figure showing all the analogs and their stepwise design. However, the author should remove the arrows between #4, 5, 6, 7, and 8, as these are derived from the last sample. Additionally, most of the information in this scheme is repetitive, as it is also shown in Table 1. Please consider rearranging for better clarity.
Answer: We suppose that this comment is about Scheme 1, and not about Figure 1. Thank you for the positive feedback on the design of Scheme 1. We appreciate your suggestion to remove arrows. However, we respectfully disagree because these are not chemical reaction arrows and #4, 5, 6, 7, and 8 are not derived from the last sample. Instead, these arrows are intended to show the logical sequence of modifications made to the parent analog (#3) and visually depict the relationships among these derivatives. Removing the arrows might diminish the clarity by isolating these analogs, potentially leading to confusion about their derivation and progression. We believe that retaining the arrows preserves the stepwise logic of the modifications and provides readers with a clear understanding of the design pathway. Regarding the overlap between Scheme 1 and Table 1, we would like to clarify that the scheme 1 serves as a visual representation of the modification strategy, which complements the detailed sequences listed in the table 1. We hope this explanation addresses your concerns and thank you for the feedback.
- For the CD results, how reliable are the structure quantifications with drug conjugates? Were the optical properties of the drug taken into consideration?
Answer: CD spectroscopy is a reliable method for studying the structure of drug conjugates. The optical properties, such as chirality and absorbance, are indeed taken into consideration during CD spectroscopy which measures the difference in the absorption of left-handed and right-handed circularly polarized light, making it highly sensitive to the chiral nature of the molecules.
The BeStSel (Beta Structure Selection) software was employed to estimate population of secondary structures of the peptides. The software is specifically designed to predict secondary structures, including parallel and antiparallel β-sheets, from CD spectra as demonstrated in multiple studies and its documentation. It achieves this by decomposing spectra into eight components, including α-helices, β-sheets (parallel and antiparallel with three degrees of twist), turns, and others, based on the Dictionary of Secondary Structure of Proteins (DSSP). Therefore, BeStSel reliably predicts parallel and antiparallel β-sheets, as supported by its validated computational framework and many published studies. For more details, kindly see the attached document, as well as a recent paper appeared in Nucleic Acids Res. 50 (2022) w90 (https://doi.org/10.1093/nar/gkac345). However, for the sake of clarity, following paragraph was added in the beginning of sub-section 2.1:
Sodium dodecyl sulfate (SDS) is commonly used to denature proteins as it disrupts non-covalent interactions. Peptides can form β-sheets depending on their hydrophobic residues, resistance to SDS-induced denaturation, helical content, and interactions with SDS micelles [61, 62]. Antiparallel refers to one of the orientations of β-sheets, wherein the strands align in opposite directions. This arrangement, along with parallel and twisted configurations, adds to the variety of structural patterns that can be observed in spectral analyses. For precise and detailed structural information derived from CD spectra, the Be-ta Structure Selection (BeStSel) method has become a widely utilized tool nowadays [63, 64].
- Parker, W. and Song, P. S., Protein structures in SDS micelle-protein complexes. Biophysical Journal,1992. 61(5): p. 1435-1439.
- Rozek, A., C.L. Friedrich, and R.E. Hancock, Structure of the bovine antimicrobial peptide indolicidin bound to dodecylphosphocholine and sodium dodecyl sulfate micelles. Biochemistry,2000. 39(51): p. 15765-15774.
- Micsonai, A., Moussong, E., Wien, F., Boros, E., Vadászi, H., Murvai, N., Lee, Y. H., Molnar, T., Refregiers, M., Goto, Y., Tantos, A., and Kardos, J. BeStSel: webserver for secondary structure and fold prediction for protein CD spectroscopy. 2022. Nucleic Acids Research, 50(W1): p. W90-W98.
- Micsonai, A., Moussong, É., Wien, F., Boros, E., Vadászi, H., Murvai, N., Lee, Y. H., Molnar, T., Refregiers, M., Goto, Y., Tantos, A., and Kardos, J. Bestsel: Updated webserver for secondary structure and fold prediction for protein CD spectroscopy. Biophysical Journal, 2023. 122(3): p. 179a.
- Figures 2-4: The image quality appears to be very low on my screen. Additionally, from the captions, it seems that the three experiments used different groups of AMP analogs. Please explain the rationale for this in the context of the study.
Answer: For figures 2–4, original high resolution AFM images have been provided now. These three figures focus on different bacterial strains (Salmonella typhi, Escherichia coli, and Pseudomonas aeruginosa) to evaluate the membrane-disrupting effects of the most active AMP analogs against each bacterium. For each strain, the three peptide analogs with the lowest MIC values were selected and tested at their double MIC concentration to ensure measurable membrane disruption. This targeted approach allowed us to obtain AFM images that effectively highlight the structural damage caused by the peptides, providing a clear visualization of their mechanism of action. We hope this explanation clarifies the rationale for using different groups of analogs in these experiments.
- The manuscript would benefit from control experiments, if not already reported, on levofloxacin's antibacterial, anti-cancer, and hemolytic properties.
Answer: Thank you for the suggestion. Data on the effects of Levo have been added in the revised version of the manuscript.
